# PARFAM—(NEURAL GUIDED) SYMBOLIC REGRESSION VIA CONTINUOUS GLOBAL OPTIMIZATION

**Philipp Scholl, Katharina Bieker, Hillary Hauger & Gitta Kutyniok**[*]
Mathematical Institute
LMU Munich
Munich, Germany
`scholl@math.lmu.de`

## ABSTRACT

The problem of symbolic regression (SR) arises in many different applications, such as identifying physical laws or deriving mathematical equations describing the behavior of financial markets from given data. In this paper, we present our new approach *ParFam* that utilizes parametric families of suitable symbolic functions to translate the discrete symbolic regression problem into a continuous one, resulting in a more straightforward setup compared to current state-of-the-art methods. In combination with a global optimizer, this approach results in a highly effective method to tackle the problem of SR. We theoretically analyze the expressivity of ParFam and demonstrate its performance with extensive numerical experiments based on the common SR benchmark suit SRBench, showing that we achieve state-of-the-art results. Moreover, we present an extension incorporating a pre-trained transformer network (*DL-ParFam*) to guide ParFam, accelerating the optimization process by up to two magnitudes. Our code and results can be found at https://github.com/Philipp238/parfam.

## 1 INTRODUCTION

Symbolic regression (SR) aims to discover concise and interpretable mathematical functions that accurately model input-output relationships. This focus on simplicity is crucial for applications requiring model analysis and trustworthiness, such as in physical or chemical sciences (Quade et al., 2016; Angelis et al., 2023; Wang et al., 2019). SR finds broad application in diverse fields, including ecosystem dynamics (Chen et al., 2019), solar power forecasting (Quade et al., 2016), financial market analysis (Liu & Guo, 2023), materials science (He & Zhang, 2021), and robotics (Oplatkova & Zelinka, 2007). The growing body of SR research, as evidenced by the increasing number of publications (Angelis et al., 2023), underscores its significance.

SR is a regression task in machine learning that aims to find an accurate model without any assumptions by the user related to the specific data set. Formally, a symbolic function $f : \mathbb{R}^n \to \mathbb{R}$ that accurately fits a given data set $(x_i, y_i)_{i=1,\ldots,N} \subseteq \mathbb{R}^n \times \mathbb{R}$ is sought, i.e., it should satisfy $y_i = f(x_i)$ for all data points, or, in the case of noise, $y_i \approx f(x_i)$ for all $i \in \{1, \ldots, N\}$. Unlike other regression tasks, SR aims at finding a simple symbolic and thus interpretable formula while assuming as little as possible about the unknown function. In contrast to SR, solutions derived via *neural networks* (NNs), for instance, lack interpretability. Traditional regression tasks, on the other hand, typically assume a strong structure of the unknown function, such as linearity or polynomial.

To tackle SR problems, the most established methods are based on *genetic programming* (Augusto & Barbosa, 2000; Schmidt & Lipson, 2009; 2010; Cranmer, 2023) and nowadays many algorithms incorporate neural networks (Martius & Lampert, 2017; Udrescu & Tegmark, 2020; Desai & Strachan, 2021; Makke et al., 2022). However, despite significant effort, many methods struggle to consistently find accurate solutions on challenging benchmarks. La Cava et al. (2021) evaluate 13 SR algorithms on the *SRBench* ground-truth problems: the Feynman (Udrescu & Tegmark, 2020)

---

[*]P. S. and G. K. are also affiliated with the Munich Center for Machine Learning (MCML), Germany. G. K. is further affiliated with the German Aerospace Center (DLR), Institute of Robotics and Mechatronics, and University of Tromsø, Norway.

and Strogatz (La Cava et al., 2016) problem sets. Both data sets consist of physical formulas with varying complexities, where the first one encompasses 115 formulas and the latter 14 ordinary differential equations. Most algorithms achieved success rates below 30% on both datasets within an 8-hour time limit, with only AI Feynman (Udrescu & Tegmark, 2020) showing better performance. Furthermore, these results deteriorate significantly in the presence of noise (La Cava et al., 2021; Cranmer, 2023).

In this paper, we introduce *ParFam*, a novel SR algorithm that leverages the inherent structure of physical formulas. By translating the discrete SR problem into a continuous optimization problem, ParFam enables precise control over the search space and facilitates the use of gradient-based optimization techniques like basin-hopping (Wales & Doye, 1997). While ParFam is not the first method to employ continuous optimization for symbolic regression, it aims to enhance the translation to the continuous space. By doing so, ParFam becomes the first SR method based on continuous optimization to achieve state-of-the-art performance. To accelerate ParFam at the cost of flexibility, we extend it to DL-ParFam which incorporates a pre-trained Set Transformer (Lee et al., 2019) to guide the optimization process for ParFam. ParFam is detailed in Section 2.1, its expressivity is analyzed in Section 2.2, and DL-ParFam is introduced in Section 2.3. Notably, despite its simplicity, ParFam achieves state-of-the-art results on the Feynman and Strogatz datasets, as demonstrated in Section 3.

**Our Contributions**    Our key contributions are as follows:

1. Introduction of ParFam, a novel method for SR improving compared to existing continuous optimization-based SR algorithm by leveraging the inherent structure of physical formulas and the expressivity of rational functions to translate SR into an efficiently solvable continuous optimization problem, by avoiding the need for nested basis functions. This results in the following advantages: (1) Enabling gradient-based optimization techniques while avoiding exploding gradients, (2) enhanced interpretability, and (3) efficient but simple and user-friendly setup.

2. Thorough theoretical analysis of the expressivity of ParFam, showing its high expressivity despite its pre-defined structure necessary for continuous optimization.

3. Introduction of DL-ParFam, an extension of ParFam based on a pre-trained Set Transformer (Lee et al., 2019), which guides ParFam and, therefore, accelerates its search by up to 100 times.

4. Extensive benchmarks showing state-of-the-art performance of ParFam and DL-ParFam and significantly better results than other methods based on continuous optimization.

**Related work**    Most SR algorithms approach the problem in two steps. First, they search for the analytic form of the target function in the discrete space of functions and then optimize the coefficients via continuous optimization techniques like BFGS (Nocedal & Wright, 2006).

Traditionally, **genetic programming** was used to heuristically search the space of equations given some base functions and operations (Augusto & Barbosa, 2000; Schmidt & Lipson, 2009; 2010; Cranmer, 2023). However, due to the accomplishments of NNs across diverse domains, numerous researchers aimed to leverage their capabilities within the realm of SR. Udrescu & Tegmark (2020), for instance, have employed an auxiliary NN to evaluate data characteristics.

Petersen et al. (2021) rely on **reinforcement learning** (RL) to explore the function space, where a policy, modeled by a recurrent neural network, generates candidate solutions. Mundhenk et al. (2021) combined this concept with genetic programming such that the RL algorithm iteratively learns to identify a good initial population for the GP algorithm, resulting in superior performance compared to individual RL and GP approaches. Similarly, Sun et al. (2022) rely on Monte Carlo tree search to search the space of expression trees for the correct equations.

Inspired by the success of **pre-training** of models on large data sets in other machine learning tasks (Kaplan et al., 2020; Devlin et al., 2018; Brown et al., 2020; Chen et al., 2020) and for mathematical data (Lample & Charton, 2019), there have been many attempts in recent years to leverage pre-training for SR. Biggio et al. (2021) build upon the data generation from Lample & Charton (2019) to train a neural network to predict the skeleton of a function symbolically, the constants of the skeleton are then found via BFGS. Kamienny et al. (2022) extend this approach to predict the skeletons

directly and only use BFGS for fine-tuning afterwards. Landajuela et al. (2022) incorporate pre-training with a combination of RL (Petersen et al., 2021), GP (Mundhenk et al., 2021), AI Feynman (Udrescu & Tegmark, 2020) and linear models.

In contrast, only a few algorithms, such as FFX (McConaghy, 2011) and SINDy (Brunton et al., 2016), share ParFam's approach of merging the search for the analytical form with coefficient optimization. These methods utilize a model with linear parameters, enabling efficient coefficient estimation via *sparse linear regression*. To expand the search space, they generate a large set of features by applying base functions to the input variables. However, this linear parameterization restricts the search space, as it cannot model non-linear parameters within the base functions.

The closest method to ParFam is EQL with division (Martius & Lampert, 2017; Sahoo et al., 2018), which makes use of **continuous optimization** and overcomes the limitation of FFX and SINDy by utilizing small NNs with $\sin$, $\cos$, and the multiplication as activation functions. The goal of EQL is to find sparse weights such that the NN reduces to an interpretable formula. However, while EQL applies linear layers between the base functions, ParFam applies rational layers. Thereby, EQL usually needs multiple layers to represent the most relevant functions, which introduces many redundancies, inflates the number of parameters, and complicates the optimization process. Moreover, EQL relies on the local minimizer ADAM (Kingma & Ba, 2014) for coefficient optimization. On the contrary, ParFam leverages the reduced dimensionality of the parameter space by applying global optimization techniques for the parameter search, which mitigates the issues of local minima. Furthermore, ParFam maintains versatility, allowing for the straightforward inclusion of different base functions, while EQL cannot handle, e.g., the exponential, logarithm, root, and division within unary operators. In recent years, several extensions of EQL and similar approaches have been proposed. DySymNet (Li et al., 2024) employs a structure akin to EQL but optimizes the architecture through reinforcement learning. MetaSymNet (Li et al., 2023) builds on the EQL framework by incorporating evolvable activation functions and rules for dynamically modifying the architecture during training. Similarly, Dong et al. (2024) enhance the classical EQL network by introducing an activation function that is evolved using genetic programming.

## 2 METHODS

In the following section, we first introduce ParFam as a novel approach, that exploits a well-suited representation of possible symbolic functions to which an efficient global optimizer can be applied. Afterwards, we theoretically analyze the expressivity of the considered parametric families, showing that ParFam is quite expressive despite the restrictions due to the predefined structure. We then introduce DL-ParFam as an extension of ParFam which incorporates pre-training on synthetic data to inform the parameter choices of ParFam and, therefore, speed up the learning process.

### 2.1 PARFAM

The aim of SR is to find a simple and thus interpretable function that describes the mapping underlying the data $(x_i, y_i)_{i=1,...,N}$ without additional assumptions. Typically, a set of base functions, such as $\{+, -, ^{-1}, \exp, \sin, \sqrt{}\}$, is predetermined. The primary goal of an SR algorithm is to find the simplest function that uses only these base functions to represent the data, where simplicity is usually defined as the number of operations. In Subsection 2.1.1 we introduce the architecture of ParFam and in Subsection 2.1.2 we explain its optimization.

### 2.1.1 THE STRUCTURE OF THE PARAMETRIC FAMILY

The main goal of ParFam is to translate SR into a continuous optimization problem to enable the use of gradient and higher-order derivative information to accelerate the search. To achieve this, we construct a parametric family of functions in a neural network-like structure presented in Figure 1, where $Q_1, ..., Q_{k+1}$ are rational functions and $g_1, ..., g_k$ are the unary base functions, which cannot be expressed as rational functions, like $\sin$, $\sqrt{}$, $\exp$, etc. The difference from a standard residual neural network with one hidden layer is that we use rational functions instead of linear connections between the layers. Furthermore, we apply physically relevant activation functions $g_1, ..., g_k$ which may differ in each neuron.

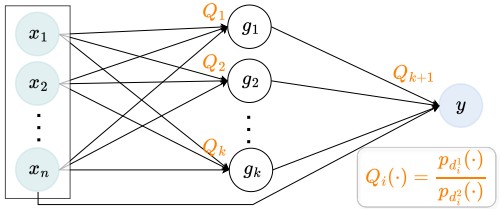

Figure 1: The architecture of ParFam: ParFam can be interpreted as a residual neural network with one hidden layer. Instead of linear weights between the layers, it applies rational functions $Q_i(\cdot) = p_{d_i^1}(\cdot)/p_{d_i^2}(\cdot)$. Furthermore, the standard basis functions are substituted by physically relevant functions like $\sin, \exp, \sqrt{}$, etc. The learnable parameters are the coefficients of $p_{d_i^1}$ and $p_{d_i^2}$.

Since the network has only one hidden layer, we can write it in a compact form as

$$f_\theta(x) = Q_{k+1}(x, g_1(Q_1(x)), g_2(Q_2(x)), \dots, g_k(Q_k(x))), \tag{1}$$

where $x \in \mathbb{R}^n$ is the input vector. The learnable parameters $\theta \in \mathbb{R}^m$ are the coefficients of the polynomials, i.e., of the numerators and denominators of $Q_1, ..., Q_{k+1}$. The degrees $d_i^1$ and $d_i^2$, $i \in \{1, \dots, k+1\}$, of the numerator and denominator polynomials of $Q_1, ..., Q_{k+1}$, respectively, and the base functions $g_1, ..., g_k$ are chosen by the user. Depending on the application, custom functions can be added to the set of base functions. This versatility and its simplicity make ParFam a highly user-friendly tool, adaptable to a wide range of problem domains. In Appendix A, we explain how to incorporate specific base functions to avoid numerical issues and further implementation details.

It is possible to extend the architecture shown in Figure 1 to multiple layers to cover arbitrary functions. However, the main motivation for the proposed architecture is that it consists of a single hidden layer, due to the high approximation qualities of rational functions and the general structure of common physical laws (Woan, 2000). Employing a single hidden layer offers several advantages: it reduces the number of parameters, simplifies optimization by mitigating issues such as exploding or vanishing gradients caused by nested functions, and enhances interpretability since it avoids complicated and uncommon compositions such as $\sin \circ \cos$ (Woan, 2000), which many algorithms enforce to avoid as well (Petersen et al., 2021; Landajuela et al., 2022). In Section 2.2 we analyze the expressivity of our architecture (using one hidden layer) and in Section 3 we show that the structure is not only flexible enough to recover many formulas exactly, but also has the best approximation capabilities among all tested algorithms.

### 2.1.2 OPTIMIZATION

The goal of the optimization is to find the coefficients of the rational functions $Q_1, ..., Q_{k+1}$ such that $f_\theta$ approximates the given data $(x_i, y_i)_{i=1,...,N}$, thus, we aim to minimize the mean squared error (MSE) between $y_i$ and $f_\theta(x_i)$. As we aim for preferably simple functions to derive interpretable and easy-to-analyze results, a regularization term $R(\theta)$ is added to encourage sparse parameters. In total, we consider the loss function

$$L(\theta) = \frac{1}{N} \sum_{i=1}^{N} (y_i - f_\theta(x_i))^2 + \lambda R(\theta), \tag{2}$$

where $\lambda > 0$ is a hyperparameter to control the weight of the regularization. Here, we choose $R(\theta) = \|\theta\|_1$ as a surrogate for the number of non-zero parameters, which is known to enforce sparsity (Bishop, 2006; Goodfellow et al., 2016). In Appendix A, we discuss how to deal with the regularization of the coefficients of rational functions in detail, to ensure that the regularization is applied effectively and cannot be circumvented during optimization.

Although the SR problem is now transformed into a continuous optimization problem, due to the presence of many local minima, it is not sufficient to apply purely local optimization algorithms like gradient descent or BFGS (Nocedal & Wright, 2006). This is also shown in our comparison study in Appendix B. To overcome these local minima, we instead rely on established (stochastic) global optimization methods. Here, we choose the so-called *basin-hopping* algorithm, originally introduced by Wales & Doye (1997), which combines a local minimizer, e.g., BFGS (Nocedal & Wright, 2006), with a global search technique inspired by Monte-Carlo minimization (Li & Scheraga, 1987) to cover a larger part of the parameter space. More precisely, we use the implementation

provided by the SciPy library (Virtanen et al., 2020). The basic idea of the algorithm is to divide the complex landscape of the loss function into multiple areas, leading to different optima. These are the so-called basins. The random perturbation of the parameters allows for hopping between these basins and the local search (based on the real loss function) inbetween improves the results and ensures that a global minimum is reached if the correct basin is chosen. For the acceptance test, the criterion introduced by Metropolis et al. (1953) is taken.

Following the optimization with basin-hopping, a finetuning routine is initiated. In this process, coefficients that fall below a certain threshold are set to 0, and the remaining coefficients are optimized using the L-BFGS method, starting from the previously found parameters. The threshold is gradually increased from $10^{-5}$ to $10^{-2}$ to encourage further sparsity in the discovered solutions. This step has been found to be crucial in enhancing the parameters initially found by basin-hopping.

## 2.2 EXPRESSIVITY OF PARFAM

Since the structure of the parametric family in ParFam restricts the search space, it is interesting to investigate the expressivity of our approach in detail. For this, we aim to quantify the ratio of the number of functions of complexity $l \in \mathbb{N}$ that can be represented using ParFam to the number of functions with the same complexity that cannot be represented. We follow the modeling approach by Lample & Charton (2019), who used expression trees to represent functions and investigate the number of expression trees, which is a common structure used in genetic algorithms. Examples of expression trees are given in Figure 2. Note that this approach does not take into account that some trees model the same expression, e.g., $x_1 + x_2$ and $x_2 + x_1$ will refer to different trees.

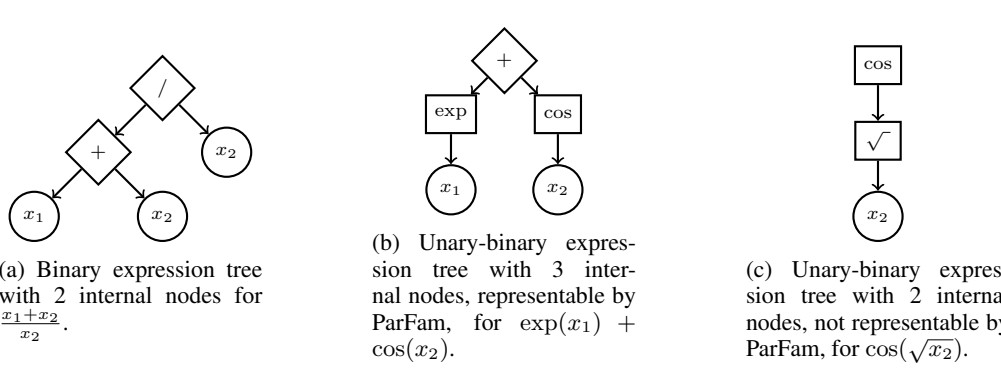

(a) Binary expression tree with 2 internal nodes for $\frac{x_1 + x_2}{x_2}$.

(b) Unary-binary expression tree with 3 internal nodes, representable by ParFam, for $\exp(x_1) + \cos(x_2)$.

(c) Unary-binary expression tree with 2 internal nodes, not representable by ParFam, for $\cos(\sqrt{x_2})$.

Figure 2: Examples for the different kinds of expression trees counted by $b_l$, $c_l$, and $d_l$.

We call a node without children a *leaf*, a node with one child a *unary node*, and a node with two children a *binary node*. The nodes with children, i.e., the unary and binary nodes, are referred to as *internal nodes*. We call a tree a *binary tree* if all its internal nodes are binary, and a *unary-binary tree* if all its internal nodes are either unary or binary. We assume that there are $n$ different possible leaves (the variables $x_1, ..., x_n$), $k$ different unary nodes (the non-rational functions $g_1, ..., g_k$), and $b$ different binary nodes (the binary operations $+, -, /, \cdot$; can be set to 4 in general). We do not model constants separately, as these can be incorporated either as part of each node or as additional leaves.

To quantify the expressivity of ParFam, we define $c_l$ as the number of unary-binary trees with $l$ internal nodes that can be represented by ParFam and $d_l$ as the number of all unary-binary trees with $l$ internal nodes (including the trees that cannot be represented by ParFam). Examples for these different types of trees are shown in Figure 2. The tree in Figure 2c cannot be modeled by ParFam since there is a path from the root (cos) to a leaf ($x_2$), that contains more than one unary node. Such paths represent compositions of unary functions $g_1, ..., g_k$ which are omitted by ParFam as explained in Section 2.1. Typical formulas from the Feynman dataset Udrescu & Tegmark (2020), for instance, have a complexity of around 10. For example, the formula $m \sin(n\theta/2)^2 / \sin(\theta/2)^2$ (Feynman I.30.3) has a complexity of 9.

Our goal in this section is to compute an estimate for the ratio $c_l/d_l$. The proofs rely mostly on the idea of generating functions (Wilf, 2005) and can be found in Appendix E together with additional context. We start by stating an approximation to $c_l$ proven in Appendix E.5.

**Theorem 2.1.** *For $(c_l)_{l \in \mathbb{N}}$, the number of unary-binary trees expressible by ParFam with complexity l, it holds that*

$$c_l = \frac{1}{2bx_1^{l+1}} \left( v_0 \left( \frac{1}{\sqrt{4\pi(l+1)^3}} + \frac{3}{8\sqrt{4\pi(l+1)^5}} \right) - v_1 \frac{3}{4\sqrt{\pi(l+1)^5}} \right) + O(x_1^{-l} l^{-7/2}) \qquad (3)$$

*with some constants $x_1, v_0, v_1 \in \mathbb{R}$ depending on the number of binary operators b, number of unary operators k, and number of variables n.*

In Appendix E.5, we additionally compute the exact formulas for $v_0$ and $v_1$. Moreover, we show the approximations and the true values $c_l$ in Figure 8a, revealing that already the first-order approximation of $c_l$ is quite close to the exact one. Since we are interested in $c_l/d_l$, we also need an approximation of $d_l$.

**Theorem 2.2.** *For $(d_l)_{l \in \mathbb{N}}$, the number of unary-binary trees with complexity l, it holds that*

$$d_l = \frac{\lambda}{2br_2^{l+1}} \left( \sqrt{1 - \frac{r_2}{r_1}} \left( \frac{1}{\sqrt{4\pi(l+1)^3}} + \frac{3}{8\sqrt{4\pi(l+1)^5}} \right) - \frac{3r_2}{8\sqrt{1 - \frac{r_2}{r_1}}\sqrt{\pi(l+1)^5 r_1}} \right) + O(x_1^{-l} l^{-7/2}) \quad (4)$$

*where $r_{1,2} = \frac{k + 2bn \pm 2\sqrt{bnk + b^2 n^2}}{k^2}$ and b, k, and n denote the number of binary operators, the number of unary operators, and the number of variables, respectively.*

The proof of Theorem 2.2 is given in Appendix E.7. Theorem 2.1 and 2.2 yield an approximation for the expressivity of ParFam $c_l/d_l$. To simplify the approximations, we disregard the constant and polynomial terms in $l$ yielding $c_l/d_l \approx (r_2/x_1)^l$. Table 1 shows $r_2/x_1$ for $b = 4$ and varying values for $k$ and $n$. The ratio is mainly between 0.9 and 1.0, revealing that especially for formulas of low complexity, i.e., small $l$, $c_l$ is relatively close to $d_l$, showing the high expressivity of ParFam despite its restrictions. For example, for $n = 4$ and $k = 3$, Table 1 yields $\frac{r_2}{x_1} = 0.9799$. Therefore, $\frac{c_l}{d_l} \approx 0.9799^l$ holds. E.g, for $l = 5$, ParFam covers 90.25% of formulas, and, for $l = 10$, 81.62%.

Table 1: Approximation of $\frac{c_{l+1}/c_l}{d_{l+1}/d_l} \approx \frac{r_2}{x_1}$, given by Theorem 2.1 and 2.2, for $b = 4$ and varying values for $k$ and $n$.

| $n/k$ | 1 | 2 | 3 | 4 | 5 | 6 |
|---|---|---|---|---|---|---|
| 1 | 0.9712 | 0.9356 | 0.9020 | 0.8713 | 0.8435 | 0.8183 |
| 2 | 0.9881 | 0.9712 | 0.9533 | 0.9356 | 0.9185 | 0.9020 |
| 3 | 0.9931 | 0.9827 | 0.9712 | 0.9593 | 0.9474 | 0.9356 |
| 4 | 0.9954 | 0.9881 | 0.9799 | 0.9712 | 0.9623 | 0.9533 |
| 5 | 0.9966 | 0.9912 | 0.9849 | 0.9782 | 0.9712 | 0.9641 |
| 6 | 0.9974 | 0.9931 | 0.9881 | 0.9827 | 0.9770 | 0.9712 |
| 7 | 0.9979 | 0.9944 | 0.9903 | 0.9859 | 0.9811 | 0.9762 |
| 8 | 0.9983 | 0.9954 | 0.9919 | 0.9881 | 0.9841 | 0.9799 |
| 9 | 0.9985 | 0.9961 | 0.9931 | 0.9899 | 0.9864 | 0.9827 |

## 2.3 DL-PARFAM

The choice of model parameters for $f_\theta$, specifically the basis functions $g_1, ..., g_k$ and the degrees of the polynomials $Q_1, ..., Q_{k+1}$, is crucial yet challenging. A highly general parametric family can lead to a complex optimization problem, while a restrictive family may hinder the discovery of the correct expression. If the time constraint allows it, iterating through different model parameters can be an effective solution for this problem, as demonstrated in our experiments in Section 3.

To solve this problem with tighter time constraints, we introduce DL-ParFam: A neural-guided version of ParFam, which follows the recent emergence of pre-trained neural networks for symbolic regression (Biggio et al., 2020; 2021; Kamienny et al., 2022; Holt et al., 2023) to predict the correct model parameters for the given data. The idea behind these approaches is to leverage the simple generation of synthetic data for symbolic regression, to train networks to map data $(x_i, y_i)_{i=1,...,N}$ directly—or indirectly using further optimization schemes— to the corresponding solution. After a costly pre-training step that can be done offline and only has to be performed once, this acquired knowledge can be used to speed up the learning process considerably.

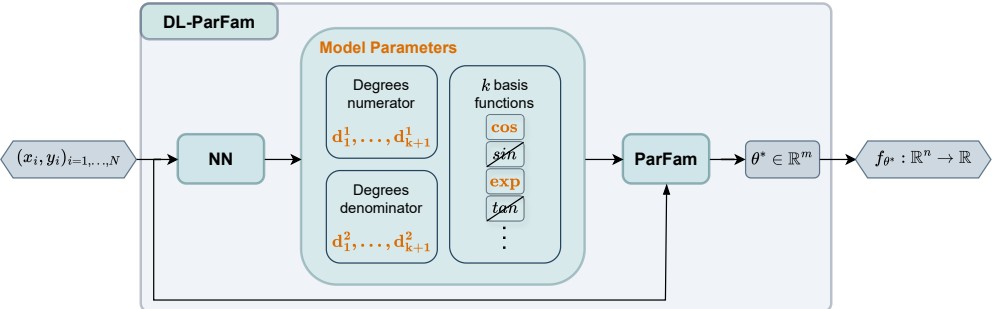

Figure 3: DL-ParFam first applies the pre-trained neural network to input data $(x_i, y_i)_{i=1,\ldots,N}$, which outputs the model parameters for ParFam: the degrees of the polynomials used in $Q_1, \ldots, Q_{k+1}$ and the basis functions $g_1, \ldots, g_k$. Afterwards, ParFam can run using these settings to find the best parameters $\theta*$ and, therefore, identify the best fitting function $f_\theta$.

However, pre-training is a highly complicated task, because of the complex data distribution and the need to be able to handle flexible data sets in high dimensions (Biggio et al., 2021; Kamienny et al., 2022). Furthermore, networks trained on the symbolic representation of functions fail to incorporate invariances in the function space during training, e.g., $x+y$ and $y+x$ are seen as different functions, as pointed out by Holt et al. (2023), which possibly complicates the training. Holt et al. (2023) resolve this by evaluating the generated function to compute the loss and update the network using RL. However, evaluating each function during the training instead of comparing its symbolic expression with the ground truth is computationally expensive. Moreover, due to the non-differentiability, the network has to be optimized using suitable algorithms like policy gradient methods. The approach most similar to DL-ParFam is SNR (Liu et al., 2023), which uses a pre-trained SET-Transformer to predict a mask for active connections within SymNet—a symbolic neural network similar to EQL. During inference, these predictions are further fine-tuned using RL.

With DL-ParFam, see Figure 3, we aim to combine the best of both worlds, by predicting the model parameters for ParFam which are more robust to functional invariances than symbolic representations. This approach yields significant improvements compared to ParFam and other pre-training-based SR methods:

- DL-ParFam strongly reduces the computational burden of ParFam by predicting model parameters.

- DL-ParFam predicts the structure of the function, which can be directly compared with the ground truth and, thereby, avoids the evaluation of the predicted function on the data in every training step and yields an end-to-end differentiable pipeline, while being able to manage invariances in the function space.

The idea behind DL-ParFam is that we train a neural network on synthetic functions to learn to predict the correct model parameters for ParFam from the data $(x_i, y_i)_{i=1,\ldots,N}$. After the training is finished, the network is used on real data to predict the model parameters, with which ParFam then aims to compute the underlying function. We implement the neural network for DL-ParFam using a Set Transformer (Lee et al., 2019) as the encoder and a fully connected ReLU network as the classifier. To train it we generate synthetic functions along with the corresponding data points $(x_i, y_i)_{i=1,\ldots,N}$ and one-hot encoded model parameters. Note that we do not follow the data generation introduced by Lample & Charton (2019) as other approaches (Kamienny et al., 2022; Biggio et al., 2021; Holt et al., 2023) do since we require direct access to the model parameters for ParFam, which would be complicated to extract from the formulas generated by Lample & Charton (2019). The $x_i$ are sampled from a uniform distribution on $[1,5]^n$, where $1 \leq n \leq 9$. The functions $f$ along with $(x_i, f(x_i) = y_i)_{i=1,\ldots,N}$ are generated as described in Appendix C. The input data to the neural network is normalized to be between $-1$ and $1$ to improve the training and make the model applicable to different input ranges. The network is trained by optimizing the cross-entropy between its output and the model parameters using the Adam optimizer with a learning rate of 0.0001 and a

step learning rate scheduler. Further implementation details and the hyper-parameters necessary to replicate the experiments are provided in Appendix D.

The network was trained on about 4,000,000 functions for 88 epochs after it stopped improving for 4 epochs. The creation of the data took 1.3h and the training took 93h on one RTX3090. We further fine-tuned the network on noisy data on 4,000,000 noisy functions for 30h on one RTX3090. For comparison, end-to-end (E2E) (Kamienny et al., 2022) made use of 800 GPUh and NeSymRes (Biggio et al., 2021), which is only trained for problems up to dimension 3, used 10,000,000 functions and was trained on a GeForce RTX 2080 GPU for 72h.

## 3 BENCHMARK

After the introduction of the SR benchmark (SRBench) by La Cava et al. (2021), several researchers have reported their findings on SRBench's ground-truth and black box data sets, due to the usage of real-world equations and data, the variety of formulas, the size of the benchmark, and comparability to competitors. The ground-truth datasets are synthetic datasets following real physical formulas and the black-box datasets are real-world datasets for which the underlying formulas are unknown. These data sets are described in more detail in Appendix F. In this section, we evaluate ParFam and DL-ParFam on the SRBench data sets and report their performance in terms of the symbolic solution rate, the coefficient of determination $R^2$, and their training time demonstrating the strong performance of ParFam and the additional speed up achieved by DL-ParFam.

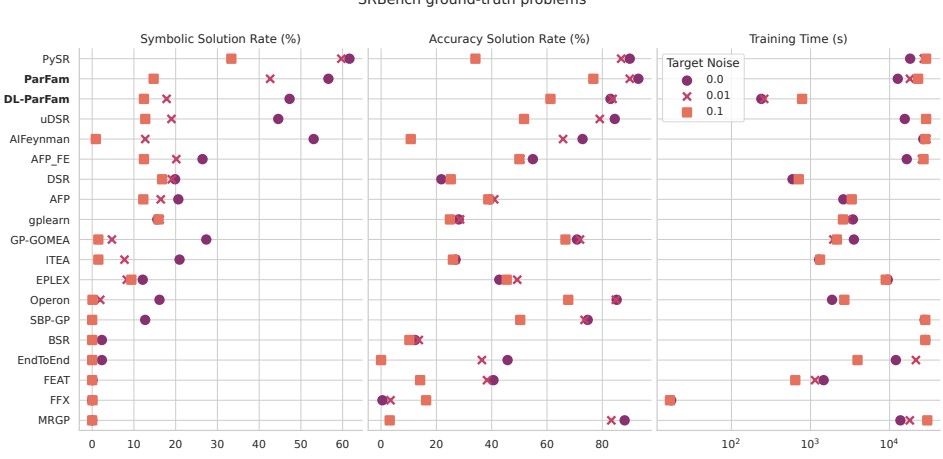

Figure 4: Mean results on the SRBench ground-truth problems. Following SRBench terminology, training time refers to the time each algorithm requires to compute a result for a specific problem, which corresponds to inference time for pre-trained methods.

**Competitors** We include the results reported by SRBench for 14 SR algorithms and extend them by running EndToEnd (Kamienny et al., 2022), uDSR (Landajuela et al., 2022), and PySR (Cranmer, 2023) on our machines. Further information on the algorithms and the chosen hyper-parameters can be found in Appendix G.

**Metrics** To ensure comparability with the results evaluated on SRBench, we use the same evaluation metrics as La Cava et al. (2021). For the ground-truth problems, we first report the symbolic solution rate, which is the percentage of equations recovered by an algorithm. Second, we consider the coefficient of determination

$$R^2 = 1 - \frac{\sum_{i=1}^{N}(y_i - \hat{y}_i)^2}{\sum_{i=1}^{N}(y_i - \bar{y})^2}, \tag{5}$$

where $\hat{y}_i = f_\theta(x_i)$ represents the model's prediction and $\bar{y}$ the mean of the output data $y$. The closer $R^2$ is to 1, the better the model describes the variation in the data. It is a widely used measure for

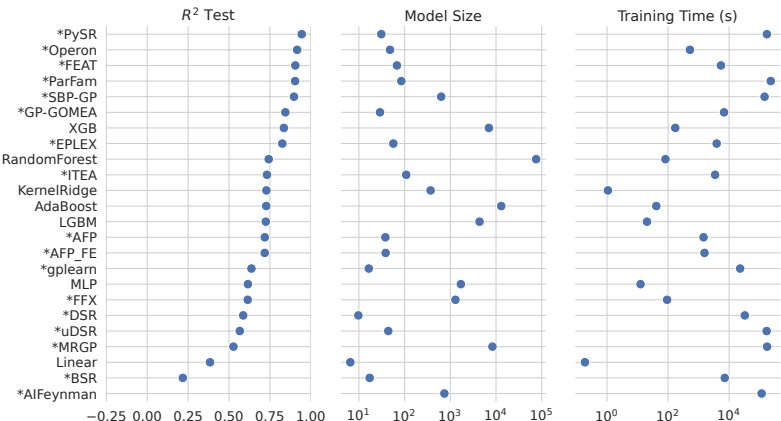

Figure 5: Median $R^2$, formula complexity, and training time on the 77 black-box problems from SRBench (La Cava et al., 2021) with at most 10 independent variables. The asterisk indicates that it is a symbolic regression method.

goodness-of-fit since it is independent of the scale and variation of the data. Following La Cava et al. (2021) we report the accuracy solution rate for each algorithm, defined as the percentage of functions such that $R^2 > 0.999$. The original data sets do not include any noise. However, similar to La Cava et al. (2021), we additionally perform experiments with noise by adding $\varepsilon_i \sim N(0, \sigma^2 \frac{1}{N} \sum_{i=1}^N y_i^2)$ to the targets $y_i$, where $\sigma$ denotes the noise level. For the black-box problems we report the median $R^2$ and the median complexity of the formula, as defined in La Cava et al. (2021), since the symbolic solution rate is not defined in this case.

**Hyperparameters** The hyperparameters of ParFam can be divided into two subsets. The first subset defines the parametric family $(f_\theta)_{\theta \in \mathbb{R}^m}$, e.g., the degree of the polynomials and the set of base functions. A good choice for this set is highly problem-dependent. However, in the absence of prior knowledge, it is advantageous to select a parametric family that is sufficiently expansive to encompass a wide range of potential functions. In this context, we opt for sin, exp, and $\sqrt{}$ as our base functions. For the first layer rationals $Q_1, \ldots, Q_k$, we set the degrees of the numerator and denominator polynomials to 2. For $Q_{k+1}$, we set the degree of the numerator polynomial to 4 and the denominator polynomial to 3. This choice results in a parametric family with hundreds of parameters, making it challenging for global optimization. To address this issue, we iterate for ParFam through various smaller parametric families, each contained in this larger family, see Appendix H for details. The second set of hyperparameters defines the optimization scheme. Here, we set the regularization parameter to $\lambda = 0.001$, the number of iterations for basin-hopping to 10, and the maximal number of BFGS steps for the local search to 100 times the dimension of the problem. Our choice of parameters for ParFam and DL-ParFam are summarized in Table 7 in Appendix I. The hyperparameters for the pre-training of DL-ParFam can be found in Appendix D.

**Results** Following La Cava et al. (2021), we allow a maximal training time of 8 CPU hours and a maximal number of function evaluations of $1,000,000$ on the ground-truth data sets and 48 CPU hours on the black-box problems. In Figure 4, we present the mean of the symbolic solution rate, the accuracy solution, and the training time on both data sets together. PySR, ParFam, AI Feynman, DL-ParFam, and uDSR outperform all other competitors by a substantial margin (over $20\%$) when it comes to symbolic solution rate. Among those 5 algorithms, PySR performs the best, followed by ParFam. These two algorithms are also the most robust to noise, where it is important to notice that PySR is the only method that incorporates a noise filter (Cranmer, 2023).

Concerning the accuracy solution, ParFam outperforms all competitors with and without noise, followed by PySR, MRGP, Operon, uDSR, and DL-ParFam. These two metrics underscore the strongly competitive performance of ParFam with the current state-of-the-art. While DL-ParFam performed slightly worse, it beats most of the established methods in both metrics, while being up to a hundred times faster than its competitors, which was the goal of incorporating pre-training into ParFam.

However, DL-ParFam's ability to recover the symbolic solution is notably hindered under low-noise conditions. The only other pre-training-based method, EndToEnd, performed worse in all three metrics. We performed the experiments without tuning the hyperparameter $\lambda$. To assess the sensitivity of the results with respect to $\lambda$, see Appendix K.

In Figure 5 we present the median $R^2$, formula complexity, and training time across the 77 black-box problems with a maximum of 10 independent variables. While ParFam's performance is slightly weaker compared to the physics datasets discussed earlier, it remains among the top-performing algorithms for these real-world datasets. Results of DL-ParFam on the black-box data sets with at most 9 variables (the limit of the current DL-ParFam) are shown in Appendix L.

Due to the proximity of EQL (Martius & Lampert, 2017; Sahoo et al., 2018) and ParFam, we deem a comparison between these two methods as highly interesting. However, the restricted expressivity of EQL makes it an unfair comparison on the whole Feynman and Strogatz dataset. For this reason, we show the results for EQL on a reduced benchmark in Appendix M. To compare DL-ParFam with further pre-training-based methods, we show the results of DL-ParFam and NeSymRes (Biggio et al., 2021) on a reduced subset of Feynman and Strogatz in Appendix N, since NeSymRes cannot handle expressions with more than 3 variables. We further compare DL-ParFam with model-parameter selection guided by Bayesian optimization in Appendix O. For results of ParFam on the Nguyen benchmark (Uy et al., 2011) and comparisons with algorithms that were not tested on SRBench, like SPL (Sun et al., 2022) and NGGP (Mundhenk et al., 2021), see Appendix P. To show the robustness of ParFam and, especially, DL-ParFam to different data domains we refer to the SRSD-Feynman benchmarks (Matsubara et al., 2024) in Appendix Q.

## 4    DISCUSSION AND CONCLUSION

This work introduces ParFam, accompanied by a theoretical analysis and extensive experiments. ParFam is the first continuous optimization-based SR algorithm to match the performance of top genetic programming methods. Furthermore, we present a novel pre-training approach that significantly outperforms existing methods and offers a substantial speed advantage over traditional competitors.

**Limitations**    The parametric structure of ParFam is its greatest asset in tackling SR but also its main constraint, as it limits the function space. However, in Section 2.2, we theoretically prove that this limitation is not severe. Section 3 further demonstrates that algorithms theoretically capable of identifying specific formulas often fail in practice, while ParFam, despite its constraints, still finds highly accurate approximations. Another drawback is the computational expense of solving high-dimensional problems ($>10$ variables) with a global optimizer, as the number of parameters grows in $O(n^d)$, where $n$ is the number of variables and $d$ the polynomial degree. For DL-ParFam, the biggest challenge is the costly pre-training of the transformer network, making it less flexible than ParFam. However, since training is done offline with synthetic data, it can be reused for various SR tasks and its pre-training is faster than other pre-training-based methods.

**Future Work**    Several avenues of ParFam and DL-ParFam remain unexplored, encompassing diverse forms of regularization, alternative parametrizations, and the potential incorporation of custom-tailored optimization techniques.

**Acknowledgement**    This work of G. K. was supported in part by the Konrad Zuse School of Excellence in Reliable AI (DAAD), the Munich Center for Machine Learning (BMBF) as well as the German Research Foundation under Grants DFG-SPP-2298, KU 1446/31-1 and KU 1446/32-1. Furthermore, P.S. and G. K. acknowledge support from LMUexcellent, funded by the Federal Ministry of Education and Research (BMBF) and the Free State of Bavaria under the Excellence Strategy of the Federal Government and the Länder as well as by the Hightech Agenda Bavaria and additional support by the project "Genius Robot" (01IS24083), funded by the Federal Ministry of Education and Research (BMBF). Moreover, this work of H. H. is supported by the DAAD programme Konrad Zuse Schools of Excellence in Artificial Intelligence, sponsored by the Federal Ministry of Education and Research.

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

# A    IMPLEMENTATION DETAILS OF PARFAM

In this section, some further implementation details are discussed.

## A.1    REGULARIZATION OF THE DENOMINATOR

Since we aim for simple function representations, i.e., for sparse solutions $\theta \in \mathbb{R}^m$, the regularization term $R(\theta)$ is of great importance. If we parameterize a rational function $Q : \mathbb{R} \to \mathbb{R}$ in one dimension by

$$Q(x) = Q_{(a,b)}(x) = \frac{\sum_{i=0}^{d^1} a_i x^i}{\sum_{i=0}^{d^2} b_i x^i} \tag{6}$$

with $a \in \mathbb{R}^{d^1+1}$ and $b \in \mathbb{R}^{d^2+1}$, the following problem occurs: Since for any $\gamma \in \mathbb{R} \setminus \{0\}$ and $(a,b) \in \mathbb{R}^{d^1+1} \times \mathbb{R}^{d^2+1}$ it holds that $Q_{(a,b)}(x) = Q_{(\gamma a, \gamma b)}(x)$, the parameters cannot be uniquely determined. Although the non-uniqueness of the solution is not a problem in itself, it shows that this parameterization is not the most efficient, and, more importantly, the regularization will be bypassed since $\gamma$ can be chosen arbitrarily small. We address this issue by normalizing the coefficients of the denominator, i.e., we use $\tilde{b} = \frac{b}{||b||_2}$ rather than $b$. In other words, instead of defining rational functions by equation 6, we consider

$$Q_{(a,b)}(x) = \frac{\sum_{i=0}^{d^1} a_i x^i}{\frac{1}{||b||_2} \sum_{i=0}^{d^2} b_i x^i}. \tag{7}$$

Note that using the 2-norm and not the 1-norm is important since we regularize the coefficients using the 1-norm. To illustrate this, let $\tilde{b} = \frac{1}{||b||_p} b$.

Case $p = 1$: When $p = 1$, we have $||\tilde{b}||_1 = 1$ for any $b \in \mathbb{R}^{d^2}$. This demonstrates that $\tilde{b}$ is not regularized anymore and, consequently, also $b$ is not regularized. In essence, this choice of $p$ does not promote sparsity in the solution.

Case $p = 2$: In contrast, when $p = 2$, we have $||\tilde{b}||_1 = ||\frac{b}{||b||_2}||_1$. This expression favors sparse solutions, as it encourages the elements of $\tilde{b}$ to be close to zero, thus promoting regularization and sparsity in the solution.

## A.2    MISCELLANEOUS

In general, we look for rational functions $Q_i$ whose numerator and denominator polynomials have a degree greater than 1 in order to model functions like $x_1^2 \exp(2x_2)$. However, for some base functions, such as $\exp, \sqrt{}, \sin, \cos$, higher powers introduce redundancy, for instance, $\exp(x_2)^2 = \exp(2x_2)$. To keep the dimension of the parameter space as small as possible without limiting the expressivity of ParFam, we allow the user to specify the highest allowed power of each chosen base function separately. In our experiments, we set it to 1 for all used basis functions: $\exp, \cos$ and $\sqrt{}$.

In addition, to ensure that the functions generated during the optimization process are always well-defined and we do not run into an overflow, we employ various strategies:

- To ensure that $\sqrt{Q(x)}$ is well-defined, i.e., $Q(x) \geq 0$ for all $x$ in the data set, we use $\sqrt{|Q(x)|}$ instead.
- To avoid the overflow that may be caused by the exponential function, we substitute it by the approximation $\min\{\exp(Q(x)), \exp(10) + |Q(x)|\}$, which keeps the interesting regime but does not run into numerical issues for big values of $Q(x)$. However, adding $|Q(x)|$ ensures that the gradient still points to a smaller $Q(x)$.
- To stabilize the division and avoid the division by 0 completely, we substitute the denominator by $10^{-5}$ if its absolute value is smaller than $10^{-5}$.

Implementing further base functions can be handled in a similar way as for the square root if they are only defined on a subset of $\mathbb{R}$ or are prone to cause numerical problems.

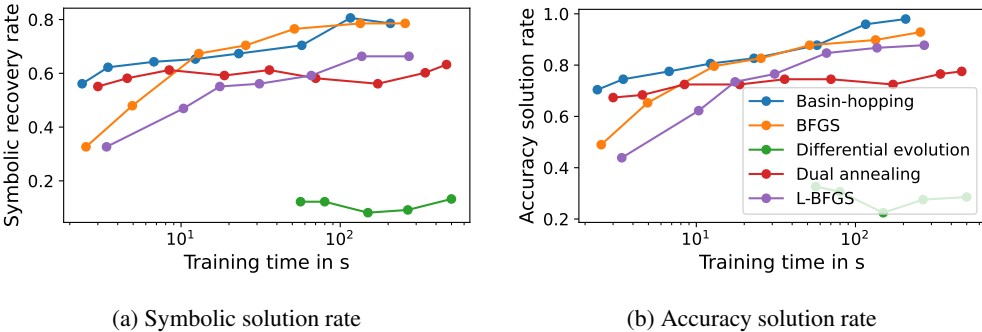

(a) Symbolic solution rate

(b) Accuracy solution rate

Figure 6: Symbolic solution and accuracy solution rate (percentage of data sets with $R^2 > 0.999$ for the test set) of ParFam with different optimizers on the subset of the Feynman problems displayed in Table 5.

## B  OPTIMIZER COMPARISON

As discussed in the main paper, ParFam needs to be coupled with a (global) optimizer to approximate the desired function. This section compares different global optimizers, underpinning our decision to use basin-hopping. We tested the following optimizers, covering different global optimizers and local optimizers combined with multi-start:

- L-BFGS with multi-start (Nocedal & Wright, 2006)
- BFGS with multi-start (Nocedal & Wright, 2006)
- Basin-hopping (Wales & Doye, 1997)
- Dual annealing (Xiang et al., 1997)
- Differential evolution (Wormington et al., 1999)

We conducted the experiments on a random subset of 15 Feynman problems, which are listed in Table 5 in Appendix F. For each of the 15 problems, we ran ParFam with each optimizer for seven different random seeds and different numbers of iterations. As we solely compare the influence of different optimizers in this experiment, we assume full knowledge of the perfect model parameters for each algorithm. Hence, we are only learning the parameters $\theta$ of one parametric family $(f_\theta)_{\theta \in \mathbb{R}^m}$ instead of iterating through multiple ones as in the experiments in Section 3. Therefore, we have to omit the problem Feynman-test-17 since the perfect model parameters result in a parametric family with too many parameters to be optimized in a reasonable time and, thus, wasting unreasonable resources. The results are presented in Figure 6. These show the superiority of basin-hopping and BFGS with multi-start compared to all the other algorithms. While basin-hopping and BFGS with multi-start perform similarly well, it is notable that basin-hopping is less sensitive to the training time (and hence the number of iterations). Therefore, we chose basin-hopping in the main paper, although using BFGS with multi-start would have led to similar results.

## C  Dataset creation for DL-ParFam

The synthetic data set for the training of DL-ParFam is sampled in the following way. For each dimension $1 \leq m \leq 9$ of $x$ and each model parameter of ParFam (i.e., maximal degree of the polynomials, base functions, etc.), we sample the same amount of functions from the parametric family $f_\theta$ by sampling $\theta \in \mathbb{R}^m$. the coefficients $\theta$, however, is not directly sampled from a specified distribution, as we have to prevent the sampling of extremely complicated and unrealistic formulas. Therefore, $\theta$ is restricted to be sparse, by limiting the number of coefficients allowed to be non-zero. Specifically, for each polynomial involved, a number between 1 and 3 is chosen, which determines the number of coefficients of the denominator and numerator polynomials that are allowed to be non-zero.

Choosing random subsets of the parameters to be non-zero, however, might produce a function that is covered by a more restrictive set of model parameters. For example, if the degree of the numerator of the output rational function $Q_{k+1}$ is supposed to be 2, but the coefficients of all monomials of degree 2 are set to 0, then the function is also covered by the "smaller" set of model parameters, with degree 1 for the numerator of $Q_{k+1}$. This hinders the training of the network since a data set can have multiple correct labels.

For this reason, we restrict the random subset of coefficients to include all coefficients that are necessary such that the generated function can not be modeled by a smaller parametric family. This process ensures that each function has as a target the "smallest" model parameter necessary to describe them and, therefore, for each input to the network there is a unique target.

Note that, it might be possible that some parts of the function can be simplified due to mathematical equivalences. However, many of these equivalences depend highly on the basis functions used, so concentrating on these would mean to overfit to specific functions which we aim to avoid.

After choosing the non-zero entries of $\theta$, we sample them from $\mathcal{N}(0, 9)$ and sample $x_1, ..., x_N \sim U(1, 5)$. The next step is to compute $y_i = f_\theta(x_i)$. As it is possible that the sampled function $f_\theta$ is not well-defined on the whole domain, some of the values $y_i$ might be NaN or $\infty$. Furthermore, we want to restrict ourselves to functions with $y_i \in [-1000, 1000]$ to ensure that the sampled functions are reasonable. Therefore, it is necessary to filter these functions afterward.

Our procedure for filtering is as follows: For 6 times we resample all $x_i$ for which $|y_i| > 1000$ or $y_i$ was NaN. This helps to keep functions that only have a singularity somewhere in the domain but are otherwise interesting. If after 6 times, there are still points in the domain with $|y_i| > 1000$ or $y_i$ NaN, we remove the generated function $f_\theta$ from the data set. In Table 2 we present a random subset of the generated expressions.

Table 2: Example formulas of the synthetically generated expressions for the pre-training of DL-ParFam

**Formula**

$$-2.665x_0^3 e^{\frac{1.0\left(-1.33x_0^2-2.51x_0+2.122\right)}{x_0}} + 4.032x_0 e^{\frac{1.0\left(-1.33x_0^2-2.51x_0+2.122\right)}{x_0}}$$

$$\frac{0.502x_0 e^{\frac{1.0(1.853x_0-0.674)}{x_0}}}{-0.887x_0^2-0.462}$$

$$\frac{3.841x_0^2 x_1+1.243x_0 \cos\left(\frac{4.048x_0}{-0.843x_0 x_1-0.537x_1}\right)}{-0.874x_0^3-0.486x_1}$$

$$-\frac{3.646x_1^2 e^{\frac{4.074x_0 x_1+0.551x_0 x_2-2.429x_1^2-0.349x_1+4.669}{-0.429x_0^2-0.574x_0 x_1+0.339x_0 x_2-0.536x_1^2-0.29x_1 x_2}}}{x_0^2}$$

$$2.226 \cos\left(\frac{0.753x_3+0.234x_4-0.735x_5}{-0.521x_0 x_4+0.853x_4 x_7}\right)$$

$$-0.392x_0 \cos\left(1.657x_0^2\right)$$

$$\frac{0.87 \cos\left(-\frac{1.0(-2.691x_0-0.831)}{x_0^2}\right)}{x_0^2}$$

$$\frac{0.297x_0}{0.669x_0^2\left(\frac{-2.898x_0-2.206}{-0.565x_0^2-0.819x_0+0.103}\right)^{0.5}+0.743\left(\frac{-2.898x_0-2.206}{-0.565x_0^2-0.819x_0+0.103}\right)^{0.5}}$$

$$\frac{1.785x_0}{-0.818x_0^2 \cos\left(\frac{0.176x_0}{0.109x_0^2+0.774x_0-0.596x_1^2+0.183x_1}\right)-0.575 \cos\left(\frac{0.176x_0}{0.109x_0^2+0.774x_0-0.596x_1^2+0.183x_1}\right)}$$

$$0.666x_1 x_2 - 1.626e^{-\frac{3.487x_2}{-0.08x_0-0.38x_1-0.785x_2-0.483}}$$

$$1.762x_1 x_2 \cos\left(\frac{1.0\left(3.598x_2^2-4.965x_3-1.02x_4 x_5\right)}{x_3}\right)$$

$$-2.773x_0^2 x_3 e^{\frac{\frac{1.482x_3 x_6}{x_1^{0.5}}}{7.804x_0^2-1.411x_0 x_2+1.884x_0 x_3-0.178x_1 x_2-2.365x_2 x_6+2.817x_2 x_7+4.086x_4^2+2.572x_5^2+1.686x_7^2}{-0.377x_0+0.128x_1-0.585x_3+0.179x_4-0.433x_5+0.054x_6-0.525}}}{0.931x_3+0.366x_7}$$

## D IMPLEMENTATION DETAILS OF DL-PARFAM

In this section, we give further details on DL-ParFam.

**Embedding** The input $(x_i, y_i)_{i=1,...,N}$ may vary in the sequence length $N$ and the dimension of $x_i \in \mathbb{R}^m$. To deal with a varying sequence length, we apply a Set Transformer (Lee et al., 2019) since the ordering of the input-output pairs is not important. To deal with the varying dimension of $x_i$, $1 \leq m \leq 9$, we embed them in $\mathbb{R}^9$. Since we noticed that a linear embedding improves the performance only marginally, we opted for a simple embedding by appending zeros.

**Target encoding** The targets of the network are the model parameters for ParFam. We focus here on five different ones: The degrees of the numerator and the denominator of the first layer rationals $Q_1, ..., Q_k$, the degrees of the numerator and the denominator of the second layer rational $Q_{k+1}$ and the functions $g_1, ..., g_k$ used. We opted for a one-hot encoding for all five parameters.

**At inference time** At inference we apply the pre-trained network to predict probabilities for the different model parameters for ParFam. We then extract $m$ sets of model parameters with the highest probability, for ParFam to try. Furthermore, we add the model parameters for polynomial and rational functions, since these involve fewer parameters and can be checked quickly. In this work, we set $m = 3$ to speed up the computations. Other works (Kamienny et al., 2022) propose to sample from the predicted probabilities to increase the diversity, which we did not consider in this work but might be a useful extension in the future.

Table 3: The training and data parameters for the pre-training of DL-ParFam

| | | |
|---|---|---|
| **Data parameters** | Maximal Degree First Layer Numerator | 2 |
| | Maximal Degree First Layer Denominator | 2 |
| | Maximal Degree Second Layer Numerator | 4 |
| | Maximal Degree Second Layer Denominator | 3 |
| | Base functions | $\sqrt{\ }$, cos, exp |
| | Maximal potence of any variable (i.e., $x_1^4$ is excluded but $x_1^3 x_2$ is allowed) | 3 |
| | Number of data pairs $(x_i, y_i)_{i=1,\dots,N}$ per function | 200 |
| | Minimal dimension of $x_i$ | 1 |
| | Maximal dimension of $x_i$ | 9 |
| **Training parameters** | Optimizer | ADAM |
| | Step size | 0.0001 |
| | Batchsize | 1024 |
| | Gradient clipping | 1 |
| | Hidden dimension | 256 |
| | Number inducing points (Set Transformer) | 128 |
| | Number heads per multi-head | 4 |
| | Number layers encoder | 8 |
| | Number layers classifier | 4 |

In Table we show the generalization gap of the SET Transformer trained for DL-ParFam, which arises by training it on synthetic data and then applying it to the Feynman data sets. We evaluated how often the SET Transformer correctly predicts model parameters for synthetic training datasets and the Feynman dataset, considering the top $k$ most likely predictions.

| | Top 1 | Top 3 | Top 5 | Top 10 |
|---|---|---|---|---|
| Synthetic | 31.4% | 50.2% | 61.8% | 71.2% |
| Feynman | 30.4% | 38.0% | 40.5% | 45.6% |

Table 4: Percentage of data sets for which the SET Transformer of DL-ParFam predicted the correct model parameters on the synthetic and the Feynman datasets.

The results indicate that while the SET Transformer used for DL-ParFam generalizes well to OOD data (the Feynman data sets) for its top predictions, there is room to optimize the synthetic training data further to improve its generalization. Note that "correct model parameters" refer to those spanning the parametric family with the minimal number of parameters covering the target function. Thus, DL-ParFam can sometimes recover the correct function without using the exact "correct" model parameters.

# E  EXPRESSIVITY OF PARFAM

Our goal in this section is to compute an estimate for the ratio $c_l/d_l$. To this end, we first consider the number of binary trees of complexity $l$, which we denote by $b_l$. Note that there exist $n$ binary trees with 0 internal nodes, thus, $b_0 = n$. A binary tree with $l$ internal nodes can be created by combining two binary trees, whose internal nodes sum up to $l-1$, with a binary operator as the root. This results in the following recurrent formula:

$$b_l = b \sum_{l_1=0}^{l-1} b_{l_1} b_{l-l_1-1}, \text{ for } l > 0, b_0 = n. \tag{8}$$

ParFam can represent any unary-binary tree where each path from the root to a leaf has at most one unary node. Thus, it holds $c_0 = n$. A new tree with $l$ internal nodes can be created by either adding a unary node as the root of a binary tree with $l-1$ internal nodes or by adding a binary node as the root of two trees that can be represented by ParFam with $l-1$ internal nodes together. Therefore,

$$c_l = kb_{l-1} + b \sum_{l_1=0}^{l-1} c_{l_1} c_{l-l_1-1}, \text{ for } l > 0,\ c_0 = n. \tag{9}$$

Lastly, we analyze the number of all unary-binary trees with $l$ internal nodes, including those not representable by ParFam. There exist $n$ unary-binary trees with 0 internal nodes, so $d_0 = n$. A new unary-binary tree with $l$ internal nodes can be created by either adding a unary node as the root of a unary-binary tree with $l-1$ internal nodes or by adding a binary node as the root of two unary-binary trees with $l-1$ internal nodes together:

$$d_l = kd_{l-1} + b \sum_{l_1=0}^{l-1} d_{l_1} d_{l-l_1-1}, \text{ for } l > 0,\ d_0 = n. \tag{10}$$

Given specific values for $k$, $b$, and $n$, we can use the formulas for $b_l$, $c_l$, and $d_l$ to calculate $c_l/d_l$ for the first few values of $l$, shown in Figure 7 in Appendix E.1. To compute the ratio $c_l/d_l$ in general, however, we need to compute an explicit exact or approximate formula for $c_l$ and $d_l$. We start by deriving an approximate formula for $c_l$. Therefore, we compute the generating function (Wilf, 2005) of $b_l$, following the ideas of Lample & Charton (2019). For the ease of notation we set $b_l = c_l = d_l = 0$ for all $l < 0$.

**Lemma E.1.** *For the generating function of $(b_l)_{l \in \mathbb{Z}}$, given by $B(z) = \sum_{l \in \mathbb{Z}} b_l z^l$, $B(z) = \frac{1-\sqrt{1-4bzn}}{2bz}$.*

The proof can be found in Appendix E.2.

Based on this formula, we can determine the generating function $C(x)$ of $(c_l)_{l \in \mathbb{Z}}$. The proof follows a similar argumentation as the one for $B(x)$ and can be found in Appendix E.3.

**Lemma E.2.** *For the generating function of $(c_l)_{l \in \mathbb{Z}}$, given by $C(z) = \sum_l c_l z^l$, $C(z) = \frac{1-\sqrt{1-4bz(kzB(z)+n)}}{2bz}$.*

To derive an approximation of $c_l$ we compute the singularity with the smallest absolute value of $C$, see Theorem 5.3.1 in Wilf (2005). The singularities of $C$ are the zeroes of

$$p(z) = 1 - 4bz(kzB(z)+n) = 1 - 2kz + 2kz\sqrt{1-4bzn} - 4bnz. \tag{11}$$

Computing a zero of $p$ is equivalent to finding a solution of

$$(1 - 2kz - 4bnz)^2 = 4k^2 z^2 (1 - 4bnz). \tag{12}$$

The solutions of this equation can be computed using Cardano's formula. Therefore, we define

$$\begin{aligned} Q &:= \frac{-4b^3 n^3 - 8b^2 k n^2 - 10bk^2 n - 3k^3}{36bk^4 n} \\ R &:= \frac{-32b^4 n^4 - 96b^3 kn^3 - 168b^2 k^2 n^2 - 140bk^3 n - 63k^4}{864bk^6 n} \end{aligned} \tag{13}$$

and

$$V := Q^3 + R^2 = -\frac{8b^2 n^2 + 13bkn + 16k^2}{27648 b^3 k^5 n^3}. \tag{14}$$

With the quantities

$$S := \sqrt[3]{R + \sqrt{V}} \quad \text{and} \quad T := \sqrt[3]{R - \sqrt{V}}, \tag{15}$$

the roots of $p$ are given by

$$\begin{aligned} x_1 &= S + T - \frac{bn+k}{3k^2} \\ x_2 &= -\frac{S+T}{2} - \frac{bn+k}{3k^2} + \frac{i\sqrt{3}}{2}(S-T) \\ x_3 &= -\frac{S+T}{2} - \frac{bn+k}{3k^2} - \frac{i\sqrt{3}}{2}(S-T). \end{aligned} \tag{16}$$

Evaluating $p$ on $x_1$, $x_2$, and $x_3$ shows that $x_1$ and $x_2$ are the zeros of $p$, if we choose the branches of the $k^{th}$-roots in $S$ and $T$ such that $\sqrt[k]{e^{i\phi}} = e^{i\phi/k}$. Choosing different branches results in the same zeros of $p$ but a different numbering. In Appendix E.4 we show that $|x_1| < |x_2|$ and, thus, $|x_1|$ is the singularity with the smallest absolute value of $C$. As discussed before, this allows us to compute an explicit approximation of $c_l$.

**Theorem E.3.** *It holds that*

$$c_l = \frac{1}{2bx_1^{l+1}} \left( v_0 \left( \frac{1}{\sqrt{4\pi(l+1)^3}} + \frac{3}{8\sqrt{4\pi(l+1)^5}} \right) -v_1 \frac{3}{4\sqrt{\pi(l+1)^5}} \right) + O(x_1^{-l}l^{-7/2}) \quad (17)$$

*for some constants $v_0, v_1 \in \mathbb{R}$ (depending on b, k, and n).*

In Appendix E.5, we additionally compute the exact formulas for $v_0$ and $v_1$. Moreover, we show the approximations and the true values $c_l$ in Figure 8a, revealing that already the first-order approximation of $c_l$ is quite close to the exact one.

Since we are interested in $\frac{c_l}{d_l}$, we also need an approximation of $d_l$. We again start by computing the generating function:

**Lemma E.4.** *For the generating function of $(d_l)_{l \in \mathbb{Z}}$, given by $D(z) = \sum_l d_l z^l$, $D(z) = \frac{1-kz-\sqrt{k^2z^2-(2k+4bn)z+1}}{2bz}$.*

The proof can be found in Appendix E.6. Lample & Charton (2019) also proved this lemma, aiming for an approximation of $d_l$ afterwards as well. However, their calculation included some small typos. For this, we derive a different approximation in Appendix E.7 and consider a higher order approximation, which is closer to $d_l$ as shown in Figure 8b.

**Theorem E.5.** *For $r_{1,2} = \frac{k + 2bn \pm 2\sqrt{bnk + b^2n^2}}{k^2}$, it holds that*

$$d_l = \frac{\lambda}{2br_2^{l+1}} \left( \sqrt{1 - \frac{r_2}{r_1}} \left( \frac{1}{\sqrt{4\pi(l+1)^3}} + \frac{3}{8\sqrt{4\pi(l+1)^5}} \right) - \frac{3r_2}{8\sqrt{1-\frac{r_2}{r_1}}\sqrt{\pi(l+1)^5 r_1}} \right) + O(x_1^{-l}l^{-7/2}). \quad (18)$$

## E.1 VISUALIZATION OF THE RATIO $c_l/d_l$

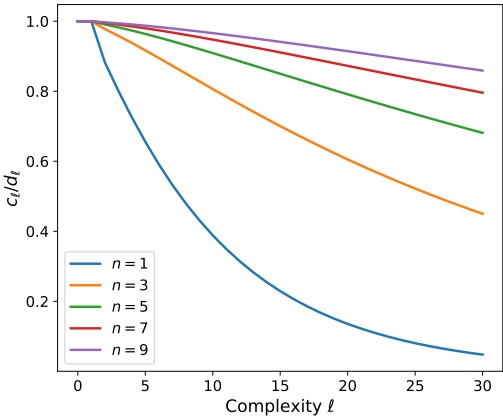

Figure 7: Ratio of $c_l$ and $d_l$ for $b = 4$ and $k = 3$ for different values of $n$ and $l = 0, 1..., 30$, computed using equation 9 and equation 10.

## E.2 PROOF OF LEMMA E.1

*Proof.* We start with multiplying equation 8 with $z^l$ and then summing over all $l \in \mathbb{Z} \setminus \{0\}$ which yields for the left hand side

$$\sum_{l \neq 0} b_l z^l = \sum_{l \in \mathbb{Z}} b_l z^l - b_0 = B(z) - n. \quad (19)$$

For the right hand side, we get

$$\sum_{l \neq 0} \left( b \sum_{l_1=0}^{l-1} b_{l_1} b_{l-l_1-1} \right) z^l = b \sum_l \left( \sum_{l_1=0}^{l-1} b_{l_1} b_{l-l_1-1} \right) z^l = b \left( \sum_l b_{l-1} z^l \right) \left( \sum_l b_l z^l \right)$$
$$= bz \left( \sum_l b_{l-1} z^{l-1} \right) \left( \sum_l b_l z^l \right) = bz B(z)^2. \tag{20}$$

Therefore,

$$bz B(z)^2 - B(z) + n = 0, \tag{21}$$

which is solved by

$$B_{1,2}(z) = \frac{1 \pm \sqrt{1 - 4bzn}}{2bz}. \tag{22}$$

Since we know that $B(0) = b_0 = n$ and $\lim_{z \downarrow 0} B_1(z) = \infty$ and $\lim_{z \to 0} B_2(z) = n$, the generating function is given by

$$B(z) = B_2(z) = \frac{1 - \sqrt{1 - 4bzn}}{2bz}. \tag{23}$$

$\square$

### E.3 PROOF OF LEMMA E.2

*Proof.* As before, we start with multiplying the recurrence relation for $c_l$ equation 9 and then sum over all $l \in \mathbb{Z} \setminus \{0\}$. For the left-hand side this yields again

$$\sum_{l \neq 0} c_l z^l = \sum_{l \in \mathbb{Z}} c_l z^l - c_0 = C(z) - n. \tag{24}$$

For the right-hand side, we get

$$k \sum_{l \neq 0} b_{l-1} z^l + b \sum_{l \neq 0} \left( \sum_{l_1=0}^{l-1} c_{l_1} c_{l-l_1-1} \right) z^l$$
$$= kz \sum_{l \neq 0} b_{l-1} z^{l-1} + b \sum_l \left( \sum_{l_1=0}^{l-1} c_{l_1} c_{l-l_1-1} \right) z^l$$
$$= kz B(z) + b \left( \sum_l c_{l-1} z^l \right) \left( \sum_l c_l z^l \right) \tag{25}$$
$$= kz B(z) + bz \left( \sum_l c_{l-1} z^{l-1} \right) \left( \sum_l c_l z^l \right)$$
$$= kz B(z) + bz C(z)^2.$$

Together this yields

$$bz C(z)^2 - C(z) + kz B(z) + n = 0, \tag{26}$$

which can be solved by

$$C_{1,2}(z) = \frac{1 \pm \sqrt{1 - 4bz(kzB(z) + n)}}{2bz}. \tag{27}$$

As before, we know that $C(0) = c_0 = n$. Therefore, $C \neq C_1$ since $\lim_{z \downarrow 0} C_1(z) = \infty$ and, thus,

$$C(z) = C_2(z) = \frac{1 - \sqrt{1 - 4bz(kzB(z) + n)}}{2bz}. \tag{28}$$

$\square$

### E.4 Proof that $|x_1| < |x_2|$

The definition of $x_1$ and $x_2$ is given in equation 16. We want to prove that $x_1$ is the singularity of $C$ with the smallest absolute value (except for $z = 0$). (We already proved in the main paper that $x_1$ and $x_2$ are the only relevant singularities.)

*Proof.* Remember that we chose the branches of the $k^{th}$-roots in $S$ and $T$ such that $\sqrt[k]{e^{i\phi}} = e^{i\phi/k}$. Start with observing that $R, V \in \mathbb{R}_{<0}$ and, therefore, $R + \sqrt{V} = \lambda e^{i\phi}$ for some $\lambda > 0$ and $\phi \in (\pi/2, \pi)$. Thus, $S = \sqrt[3]{\lambda}e^{i\phi/3}$ and $\phi/3 \in (\pi/6, \pi/3)$, so $Re(S), Im(s) > 0$ and $Im(s) > \tan(\pi/6)Re(S)$. Next observe that $T$ is the complex conjugate of $S$ and, therefore, $S+T = 2Re(S)$ and $S - T = i2Im(S)$. All together this yields

$$
\begin{aligned}
|x_2| &= \frac{S+T}{2} + \frac{bn+k}{3k^2} - \frac{i\sqrt{3}}{2}(S-T) \\
&= Re(S) + \frac{bn+k}{3k^2} - \frac{i\sqrt{3}}{2}(i2Im(S)) \\
&= Re(S) + \frac{bn+k}{3k^2} + \sqrt{3}Im(S) \\
&> Re(S) + \frac{bn+k}{3k^2} + \sqrt{3}\tan(\pi/6)Re(S) \\
&= Re(S) + \frac{bn+k}{3k^2} + \sqrt{3}\frac{1}{\sqrt{3}}Re(S) \\
&= 2Re(S) + \frac{bn+k}{3k^2} \\
&> |2Re(s) - \frac{bn+k}{3k^2}| = |x_1|.
\end{aligned}
\tag{29}
$$

$\square$

### E.5 Proof of Theorem 2.1

*Proof.* Our plan is to use Theorem 5.3.1 in Wilf (2005) stating that

$$
[z^l]\{(1 - z/s)^\beta v(z)\} = \sum_{j=0}^{m} v_j s^{-l} \binom{l - \beta - j - 1}{l} + O(s^{-l}l^{-m-\beta-2}),
\tag{30}
$$

where $[z^l]f(z)$ denotes the $l$-th coefficient (corresponding to $z^l$) of the series of powers of $f$ in $z$ and $v(z) = \sum_{j=0}^{\infty} v_j(1 - z/s)^j$ is analytic on a disk $|z| < |s| + \eta$ for some $\eta > 0$ and $\beta \notin \mathbb{N}$ and $m \in \mathbb{N}$. We define

$$
R(z) = \sqrt{1 - 2kz + 2kz\sqrt{1 - 4bzn} - 4bnz}
\tag{31}
$$

such that

$$
C(z) = \frac{1 - R(z)}{2bz}.
\tag{32}
$$

Hence, the coefficients of the power series of $R$ will allow us to compute the coefficients of the power series of $C$. To compute the power series of $R$ we set

$$
v(z) = (1 - \frac{z}{x_1})^{-1/2}R(z).
\tag{33}
$$

Since $R$ only has the two singularities $x_1$ and $x_2$ and the product of two functions which are analytic at a point $z_0$ is itself analytic at $z_0$ we know that $v$ has at most the singularities $x_1$ and $x_2$. To see

that $v$ is analytic around $x_1$, calculate

$$v(z) = (1 - \frac{z}{x_1})^{-1/2}\sqrt{1 - 2kz + 2kz\sqrt{1 - 4bzn} - 4bnz} \tag{34}$$

$$= (1 - \frac{z}{x_1})^{-1/2}\frac{\sqrt{(1 - 2kz - 4bnz + 2kz\sqrt{1 - 4bzn})(1 - 2kz - 4bnz - 2kz\sqrt{1 - 4bzn})}}{\sqrt{(1 - 2kz - 4bnz - 2kz\sqrt{1 - 4bzn})}} \tag{35}$$

$$= (1 - \frac{z}{x_1})^{-1/2}\frac{\sqrt{(1 - 2kz - 4bnz)^2 - (2kz)^2(1 - 4bzn)}}{\sqrt{(1 - 2kz - 4bnz - 2kz\sqrt{1 - 4bzn})}}. \tag{36}$$

The numerator in the last step is the same polynomial arising in equation 11 and, thus, has the roots $x_1$, $x_2$, and $x_3$. Therefore, we can factorize it and continue the computation to

$$v(z) = (1 - \frac{z}{x_1})^{-1/2}\frac{\sqrt{\mu(1 - \frac{z}{x_1})(1 - \frac{z}{x_2})(1 - \frac{z}{x_3})}}{\sqrt{(1 - 2kz - 4bnz - 2kz\sqrt{1 - 4bzn})}} \tag{37}$$

$$= \frac{\sqrt{\mu(1 - \frac{z}{x_2})(1 - \frac{z}{x_3})}}{\sqrt{(1 - 2kz - 4bnz - 2kz\sqrt{1 - 4bzn})}} \tag{38}$$

with $\mu = -16k^2bnx_1x_2x_3$. Since $(1 - 2kz - 4bnz - 2kz\sqrt{1 - 4bzn})$ has no zero at $x_1$, this shows that $v$ is analytic in $x_1$ and, thus, $x_2$ is the only singularity of $v$. Furthermore, since we have shown in Appendix E.4 that $|x_1| < |x_2|$ we know that there is some $\eta > 0$ such that $v$ is analytic on some disk $|z| < |x_1| + \eta$. Thus, we can use equation 30 to get

$$[z^l]R(z) = v_0 x_1^{-l}\binom{l - 1/2 - 1}{l} + v_1 x_1^{-l}\binom{l - 3/2 - 1}{l} + O(x_1^{-l}l^{-7/2}). \tag{39}$$

First we need to determine $v_0$ and $v_1$ which we can compute by developing $v$ around $x_1$:

$$v(z) \approx v(x_1) + v'(x_1)(z - x_1) = v(x_1) - x_1v'(x_1)(1 - z/x_1). \tag{40}$$

So, we compute

$$v_0 = v(x_1) = \frac{\sqrt{\mu(1 - \frac{x_1}{x_2})(1 - \frac{x_1}{x_3})}}{\sqrt{(1 - 2kx_1 - 4bnx_1 - 2kx_1\sqrt{1 - 4bx_1n})}} \tag{41}$$

and

$$v_1 = - x_1v'(x_1) \tag{42}$$

$$= - x_1\frac{\mu}{2\sqrt{\mu(1 - \frac{x_1}{x_2})(1 - \frac{x_1}{x_3})}\sqrt{(1 - 2kx_1 - 4bnx_1 - 2kx_1\sqrt{1 - 4bx_1n})}}(\frac{2x_1}{x_2x_3} - \frac{1}{x_2} - \frac{1}{x_3}) \tag{43}$$

$$+ x_1\frac{\sqrt{\mu(1 - \frac{x_1}{x_2})(1 - \frac{x_1}{x_3})}}{2(1 - 2kx_1 - 4bnx_1 - 2kx_1\sqrt{1 - 4bx_1n})^{3/2}} \tag{44}$$

$$\cdot (-2k - 4bn - 2k\sqrt{1 - 4bx_1n} + \frac{4bnkx_1}{\sqrt{1 - 4bx_1n}}). \tag{45}$$

Furthermore, we can use the formula (Wilf, 2005)

$$\binom{l - \alpha - 1}{l} = \frac{l^{-\alpha-1}}{\Gamma(-\alpha)}[1 + \frac{\alpha(\alpha + 1)}{2l} + O(l^{-2})] \tag{46}$$

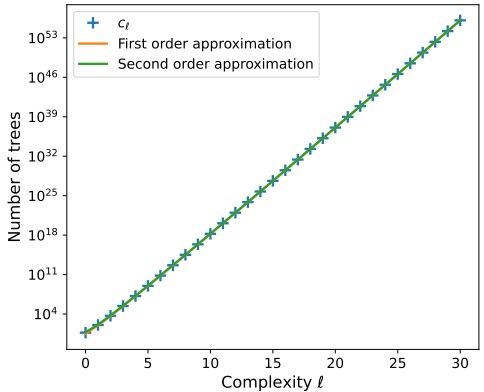 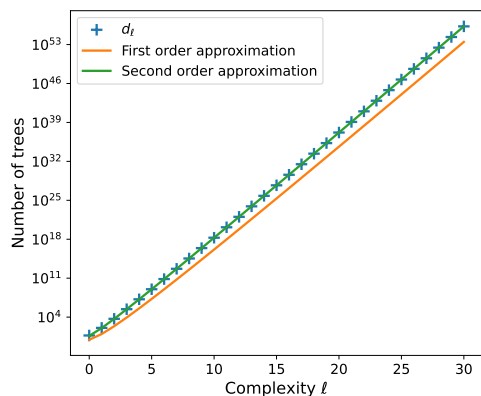

(a) The true values of $c_l$ and its approximations computed using equation 50 (first order approximation) and equation 51 (second order approximation) for $b = 4$, $k = 3$, and $n = 5$.

(b) The true values of $c_l$ and its approximations computed using equation 68 (first order approximation) and equation 69 (second order approximation) for $b = 4$, $k = 3$, and $n = 5$.

Figure 8: The true values of $c_l$ and of $d_l$ and their approximations computed in Theorem 2.1 and 2.2 for $b = 4$, $k = 3$, and $n = 5$.

to compute

$$\binom{l - 1/2 - 1}{l} = \frac{l^{-3/2}}{\Gamma(-1/2)}[1 + \frac{1/2 \cdot (1/2 + 1)}{2l} + O(l^{-2})] = -\frac{1}{\sqrt{4\pi l^3}} - \frac{3}{8\sqrt{4\pi l^5}} + O(l^{-7/2}) \tag{47}$$

and

$$\binom{l - 3/2 - 1}{l} = \frac{l^{-5/2}}{\Gamma(-3/2)}[1 + O(l^{-1})] = \frac{3}{4\sqrt{\pi l^5}} + O(l^{-7/2}) \tag{48}$$

where we used that $\Gamma(-1/2) = -\sqrt{4\pi}$ and $\Gamma(-3/2) = \frac{4\sqrt{\pi}}{3}$.

Using the approximation from equation 47 and equation 48 yields

$$[z^l]R(z) = x_1^{-l}v_0(-\frac{1}{\sqrt{4\pi l^3}} - \frac{3}{8\sqrt{4\pi l^5}}) + x_1^{-l-1}v_1\frac{3}{4\sqrt{\pi l^5}} + O(r_2^{-l}l^{-7/2})$$
$$= -x_1^{-l}v_0\frac{1}{\sqrt{4\pi l^3}} + O(r_2^{-l}l^{-5/2}) \tag{49}$$

We can now further compute, for $l > 0$,

$$c_l = [z^l]C(z) = -\frac{1}{2b}[z^l]\frac{R(z)}{z} = -\frac{1}{2b}[z^{l+1}]R(z) = x_1^{-l-1}v_0\frac{1}{2b\sqrt{4\pi(l+1)^3}} + O(x_1^{-l}l^{-5/2}) \tag{50}$$

or

$$c_l = -\frac{1}{2b}[z^{l+1}]R(z) \tag{51}$$

$$= \frac{1}{2b}\left(x_1^{-l-1}v_0(\frac{1}{\sqrt{4\pi(l+1)^3}} + \frac{3}{8\sqrt{4\pi(l+1)^5}}) - x_1^{-l-1}v_1\frac{3}{4\sqrt{\pi(l+1)^5}}\right) + O(x_1^{-l}l^{-7/2}). \tag{52}$$

$\square$

### E.6   PROOF OF LEMMA E.4

*Proof.* As for $b_l$ and $c_l$ we get for the left-hand side

$$\sum_{l \neq 0} d_l z^l = \sum_{l \in \mathbb{Z}} d_l z^l - d_0 = D(z) - n \tag{53}$$

and for the right-hand side

$$k \sum_{l \neq 0} d_{l-1} z^l + b \sum_{l \neq 0} \left( \sum_{l_1=0}^{l-1} d_{l_1} d_{l-l_1-1} \right) z^l$$

$$= kz \sum_l d_{l-1} z^l + b \sum_l \left( \sum_{l_1=0}^{l-1} d_{l_1} d_{l-l_1-1} \right) z^l$$

$$= kz \sum_l d_{l-1} z^{l-1} + b \left( \sum_l d_{l-1} z^l \right) \left( \sum_l d_l z^l \right) \tag{54}$$

$$= kzD(z) + bz \left( \sum_l d_{l-1} z^{l-1} \right) \left( \sum_l d_l z^l \right)$$

$$= kzD(z) + bzD(z)^2.$$

Together this yields

$$bzD(z)^2 + (kz - 1)D(z) + n = 0 \tag{55}$$

which is solved by

$$D_{1,2}(z) = \frac{-(kz-1) \pm \sqrt{(kz-1)^2 - 4bzn}}{2bz}. \tag{56}$$

As before, using $D(0) = n$ yields

$$D(z) = D_2(z) = \frac{-(kz-1) - \sqrt{k^2 z^2 - (2k + 4bn)z + 1}}{2bz}. \tag{57}$$

$\square$

### E.7 PROOF OF THEOREM 2.2

As for Theorem 2.1 we plan to use Theorem 5.3.1 in Wilf (2005) which yields that

$$[z^l]\{(1 - z/s)^\beta v(z)\} = \sum_{j=0}^m v_j s^{-l} \binom{l - \beta - j - 1}{l} + O(s^{-l} l^{-m-\beta-2}), \tag{58}$$

where $[z^l] f(z)$ denotes the $l$-th coefficient (corresponding to $z^l$) of the series of powers of $f$ in $z$ and $v(z) = \sum_{j=0}^\infty v_j (1 - z/s)^j$ is analytic on a disk $|z| < |s| + \eta$ for some $\eta > 0$ and $\beta \notin \mathbb{N}$ and $m \in \mathbb{N}$. Similar to the proof of Theorem 2.1 in Appendix E.5 we start by defining

$$R(z) = \sqrt{k^2 z^2 - (2k + 4bn)z + 1} \tag{59}$$

such that

$$D(z) = \frac{1 - kz - R(z)}{2bz}. \tag{60}$$

To be able to define $v$ we first have to compute the singularities of $R$ with the smallest absolute value. The singularities of $R$ are the zeros of $p(z) = k^2 z^2 - (2k + 4bn)z + 1$ which are

$$r_{1,2} = \frac{2k + 4bn \pm \sqrt{(2k + 4bn)^2 - 4k^2}}{2k^2} = \frac{k + 2bn \pm \sqrt{(k + 2bn)^2 - k^2}}{k^2}. \tag{61}$$

Since $k + 2bn > 0$ and $(k + 2bn)^2 - k^2 > 0$, we know that $|r_1| > |r_2|$ for all $b, k, n > 0$. Thus, we define

$$v(z) = (1 - \frac{z}{r_2})^{-1/2} R(z) = \lambda \sqrt{1 - \frac{z}{r_1}} \tag{62}$$

where $\lambda = k\sqrt{r_1 r_2}$ which is analytic on a disk $|z| < |r_2| + \eta$ for some $\eta > 0$ since $|r_1| > |r_2|$. Choosing $m = 1$ in equation 58, we only need to compute $v_0$ and $v_1$ which we can do by developing $v$ around $r_2$:

$$v(z) \approx v(r_2) + v'(r_2)(z - r_2) = v(r_2) - r_2 v'(r_2)(1 - z/r_2) \tag{63}$$

which shows that

$$v_0 = v(r_2) = \lambda\sqrt{1 - r_2/r_1} \tag{64}$$

and

$$v_1 = -r_2 v'(r_2) = \frac{\lambda r_2}{2\sqrt{1 - \frac{r_2}{r_1}} r_1} \tag{65}$$

since

$$v'(z) = -\frac{\lambda}{2\sqrt{1 - \frac{z}{r_1}} r_1}. \tag{66}$$

Plugging $v_0$ and $v_1$ and the approximations for the binomial coefficients from equation 47 and equation 48 into equation 58 yields

$$[z^l]R(z) = r_2^{-l}\lambda\sqrt{1 - r_2/r_1}\left(-\frac{1}{\sqrt{4\pi l^3}} - \frac{3}{8\sqrt{4\pi l^5}}\right) + r_2^{-l}\frac{\lambda r_2}{2\sqrt{1 - \frac{r_2}{r_1}} r_1}\frac{3}{4\sqrt{\pi l^5}} + O(r_2^{-l}l^{-7/2})$$

$$= -r_2^{-l}\lambda\sqrt{1 - r_2/r_1}\frac{1}{\sqrt{4\pi l^3}} + O(r_2^{-l}l^{-5/2}). \tag{67}$$

We can now further compute, for $l > 1$,

$$d_l = [z^l]D(z) = -\frac{1}{2b}[z^l]\frac{R(z)}{z} = -\frac{1}{2b}[z^{l+1}]R(x) \approx r_2^{-l-1}\lambda\sqrt{1 - r_2/r_1}\frac{1}{2b\sqrt{4\pi(l+1)^3}}. \tag{68}$$

or

$$d_l = -\frac{1}{2b}[z^{l+1}]R(x) \tag{69}$$

$$\approx \frac{1}{2b}\left(r_2^{-l-1}\lambda\sqrt{1 - \frac{r_2}{r_1}}\left(\frac{1}{\sqrt{4\pi(l+1)^3}} + \frac{3}{8\sqrt{4\pi(l+1)^5}}\right) - r_2^{-l-1}\frac{\lambda r_2}{2\sqrt{1-\frac{r_2}{r_1}} r_1}\frac{3}{4\sqrt{\pi(l+1)^5}}\right) \tag{70}$$

which perfectly approximates $d_l$.

## F  SRBench datasets

SRBench (La Cava et al., 2021) comprises two types of problem sets: the ground-truth datasets, which include the Feynman datasets (Udrescu & Tegmark, 2020) and the Strogatz datasets (La Cava et al., 2016), and the black-box datasets. The ground-truth datasets are synthetically generated and follow known analytical formulas, while the black-box datasets consist of real-world data, where the existence of concise analytic formulas is uncertain, and they are often high-dimensional. In the following, we describe the different data sets in more detail.

### F.1  Feynman data set

The Feynman data set consists of 119 physical formulas taken from the Feynman lectures and other seminal physics books (Udrescu & Tegmark, 2020). Some examples can be found in Appendix F. The formulas depend on a maximum of 9 independent variables and are composed of the elementary functions $+, -, *, /, \sqrt{}, \exp, \log, \sin, \cos, \tanh, \arcsin$ and $\arccos$. Following La Cava et al. (2021), we omit three formulas containing $\arcsin$ and $\arccos$ and one data set where the ground-truth formula is missing. Additionally, since the data sets contain more data points than required for the reconstruction of the equations and this abundance of data slows down the optimizer, we only consider a subset of 500, for the experiments without noise, and 1,000, for the experiments with noise, data points of the training data for each problem. We only use the full data sets for EndToEnd (Kamienny et al., 2022) to perform the bagging described in their paper.

Table 5 shows a random subset of the Feynman data set. The complete Feynman data set can be downloaded from the Penn Machine Learning Benchmarks (MIT license).

Table 5: Random subset of 15 equations of the Feynman problem set (Udrescu & Tegmark, 2020).

| Name | Formula |
|---|---|
| Feynman-III-4-33 | $y = \dfrac{h\omega}{2\pi \left(\exp\left(\frac{h\omega}{2\pi Tkb}\right) - 1\right)}$ |
| Feynman-III-8-54 | $y = \sin^2\left(\dfrac{2\pi E_n t}{h}\right)$ |
| Feynman-II-15-4 | $y = -Bmom\cos(\theta)$ |
| Feynman-II-24-17 | $y = \sqrt{-\dfrac{\pi^2}{d^2} + \dfrac{\omega^2}{c^2}}$ |
| Feynman-II-34-29b | $y = \dfrac{2\pi BJzgmom}{h}$ |
| Feynman-I-12-5 | $y = Efq_2$ |
| Feynman-I-18-4 | $y = \dfrac{m_1 r_1 + m_2 r_2}{m_1 + m_2}$ |
| Feynman-I-38-12 | $y = \dfrac{\varepsilon h^2}{\pi m q^2}$ |
| Feynman-I-39-22 | $y = \dfrac{Tkbn}{V}$ |
| Feynman-I-40-1 | $y = n_0 \exp\left(-\dfrac{gmx}{Tkb}\right)$ |
| Feynman-I-43-31 | $y = Tkbmob$ |
| Feynman-I-8-14 | $y = \sqrt{(-x_1 + x_2)^2 + (-y_1 + y_2)^2}$ |
| Feynman-I-9-18 | $y = \dfrac{Gm_1 m_2}{(-x_1 + x_2)^2 + (-y_1 + y_2)^2 + (-z_1 + z_2)^2}$ |
| Feynman-test-17 | $y = \dfrac{m^2\omega^2 x^2\left(\frac{\alpha x}{y} + 1\right) + p^2}{2m}$ |
| Feynman-test-18 | $y = \dfrac{3\left(H_G^2 + \frac{c^2 k_f}{r^2}\right)}{8\pi G}$ |

## F.2 STROGATZ DATA SET

The Strogatz data set introduced by La Cava et al. (2016) is the second ground-truth problem set included in SRBench (La Cava et al., 2021). It consists of 14 non-linear differential equations describing seven chaotic dynamic systems in two dimensions, listed in Appendix F.2. Each problem set contains 400 samples.

Table 6 shows the complete Strogatz data set. It can be downloaded from the Penn Machine Learning Benchmarks (MIT license).

Table 6: The Strogatz ODE problem set (La Cava et al., 2016).

| Name | Formula |
|---|---|
| Bacterial Respiration | $\dot{x} = -\frac{xy}{0.5x^2+1} - x + 20$ |
| | $\dot{y} = -\frac{xy}{0.5x^2+1} + 10$ |
| Bar Magnets | $\dot{x} = -\sin(x) + 0.5\sin(x-y)$ |
| | $\dot{y} = -\sin(y) - 0.5\sin(x-y)$ |
| Glider | $\dot{x} = -0.05x^2 - \sin(y)$ |
| | $\dot{y} = x - \frac{\cos(y)}{x}$ |
| Lotka-Volterra interspecies dynamics | $\dot{x} = -x^2 - 2xy + 3x$ |
| | $\dot{y} = -xy - y^2 + 2y$ |
| Predator Prey | $\dot{x} = x\left(-x - \frac{y}{x+1} + 4\right)$ |
| | $\dot{y} = y\left(\frac{x}{x+1} - 0.075y\right)$ |
| Shear Flow | $\dot{x} = \cos(x)\cot(y)$ |
| | $\dot{y} = \left(0.1\sin^2(y) + \cos^2(y)\right)\sin(x)$ |
| van der Pol oscillator | $\dot{x} = -\frac{10x^3}{3} + \frac{10x}{3} + 10y$ |
| | $\dot{y} = -\frac{x}{10}$ |

## F.3 BLACK-BOX DATA SETS

The black-box data sets comprise 133 complicated, real-world data sets for which the underlying formula if there even is one, is unknown. The problems are often high-dimensional. As mentioned in the limitations, ParFam is not yet suited for extremely high-dimensional datasets, so we restrict our focus to black-box datasets with a dimensionality of 10 or fewer, which leaves us with 77 out of the total 122 datasets.

## G BENCHMARKED ALGORITHMS

For our experiments on the Feynman (Udrescu & Tegmark, 2020) and the Strogatz (La Cava et al., 2016) datasets we decided to include—besides the algorithms benchmarked in La Cava et al. (2021)—algorithms which showed state-of-the-art performance in other experiments and pre-training based methods which uploaded their code and weights to have good comparisons for ParFam and DL-ParFam.

**PySR** PySR (Cranmer, 2023) uses a multi-population evolutionary algorithm, which consists of a unique evolve-simplify-optimize loop as its search algorithm. Furthermore, it involves a noise

filter to become more robust and applicable to real data. In the experiments, we used 6 populations of size 50 and 500 cycles per iteration. The formulas are restricted to have a maximum depth of 10 and a maximum size of 50. These are the default parameters shown in the tutorial, which also performed the best on a grid hyperparameter search on the first 20 problems of the Feynman dataset. Our experiments use the PySR package version 0.18.1 (Apache license 2.0).

**uDSR**  uDSR (Landajuela et al., 2022) is a combination of the recursive problem simplification introduced in Udrescu & Tegmark (2020), DSR (Petersen et al., 2021), genetic programming, pre-training, and linear models. It can, therefore, be seen as the extension of the combination of DSR and genetic programming introduced in Mundhenk et al. (2021). The experiments Landajuela et al. (2022) performed in their paper on the Feynman and Strogatz were limited to 24CPUh and 2,000,000 evaluations. For this reason, we reran uDSR on our machines with the limits specified by La Cava et al. (2021): 8CPUh and 1,000,000. Because of this, the performance we report is slightly worse. We performed the same hyperparameter search as explained for PySR and picked, in the end, their default hyperparameters (specified in the config_regression_gp.json commit: 2069d4e). Our experiments use the official DSO github repository (BSD 3-Clause License).

**EndToEnd**  EndToEnd (Kamienny et al., 2022) utilizes a pre-trained transformer to predict the symbolic form of the formula directly with estimations of the constants as well. To the best of our knowledge, this is the only pre-trained method that is able to handle 9 dimensions and arbitrary constants—which is necessary for the Feynman dataset—for which the model weights are available. Instead of the hyperparameters specified in the paper (100 bagging bags, 10 expression trees per bag, and 10 trees that are refined) we use 500 bagging bags, 10 expression trees per bag, and 100 trees that are refined since we experienced a better performance with these. Our experiments use the official github repository (Apache 2.0 license).

**NeSymRes**  NeSymRes (Biggio et al., 2021) works similarly to EndToEnd (Kamienny et al., 2022), however, they only predict the skeleton of the expression, leaving placeholders for all the constants. These are learned afterward using BFGS. Since NeSymRes only works for dimensions 1 to 3, we run it on a restricted subset of the Feynman and Strogatz datasets, the results are shown in Appendix N. Since we experienced the best performance with them, we performed the experiments with the hyperparameters setting specified in Biggio et al. (2021): beam size of 32 and 4 restarts. Our experiments use the official github repository (MIT license).

## H  MODEL PARAMETER SEARCH FOR PARFAM

The success of ParFam depends strongly on a good choice of the model parameters: The set of base functions $g_1, ..., g_k$ and the degrees $d_i^1$ and $d_i^2$, $i \in \{1, ..., k+1\}$, of the numerator and denominator polynomials of $Q_1, ..., Q_{k+1}$, respectively. On the one hand, choosing the degrees very small or the set of base functions narrow might restrict the expressivity of ParFam too strongly and exclude the target function from its search space. On the other hand, choosing the degrees too high or a very broad set of base functions can yield a search space that is too high-dimensional to be efficiently handled by a global optimization method. This might prevent ParFam from identifying even very simple functions.

To balance this tradeoff, we allow ParFam to iterate through many different choices for the hyperparameters describing the model. The user specifies upper bounds on the degrees $d_i^1$ and $d_i^2$ of the polynomials and the set of base functions $g_1, \ldots, g_k$. ParFam then automatically traverses through different settings, starting from simple polynomials to rational functions to more complex structures involving the base functions and ascending degrees of the polynomials. The exact procedure is shown in Algorithm 1. Note that we refer to the rational functions $Q_1, ..., Q_k$, which will be the inputs to the base functions, as the 'input rationals' and, therefore, describe the degrees of their polynomials by 'DegInputNumerator' and 'DegInputDenominator'. Similarly, we denote the degrees of the polynomials of the output rational function $Q_{k+1}$ by 'DegOutputNumerator' and 'DegOutputDenominator'.

---

**Algorithm 1:** Traversal of the model parameters

---

**Input:** Maximal Degree Input Numerator $d^1_{\mathsf{max,in}}$,
       Maximal Degree Output Numerator $d^1_{\mathsf{max,out}}$,
       Maximal Degree Input Denominator $d^2_{\mathsf{max,in}}$,
       Maximal Degree Output Denominator $d^2_{\mathsf{max,out}}$,
       Maximal number of base functions $b_{\mathsf{max}}$
       Set of base functions $G_{\mathsf{max}} = \{g_1, \ldots, g_k\}$.
**Output:** List of model parameters $\mathcal{L}$ that define the models ParFam can iterate through.

---

1 Let $\mathcal{L} = \{\ \}$ be an empty list.
   // Start with a polynomial model:
2 $D_{\mathsf{p}}$ = {'DegInputNumerator': 0, 'DegOutputNumerator': $d^1_{\mathsf{max,out}}$, 'DegInputDenominator': 0, 'DegOutputDenominator': 0, 'baseFunctions': []}
3 $\mathcal{L}$.append($D_0$)
   // Continue with purely rational models with different degrees:
4 **for** $d^2_{\mathsf{out}} = 1$ **to** $d^2_{\mathsf{max,out}}$ **do**
5    **for** $d^1_{\mathsf{out}} = 1$ **to** $d^1_{\mathsf{max,out}}$ **do**
6       $D_{\mathsf{r}}$ = {'DegInputNumerator': 0, 'DegOutputNumerator': $d^1_{\mathsf{out}}$, 'DegInputDenominator': 0, 'DegOutputDenominator': $d^2_{\mathsf{out}}$, 'baseFunctions': []}
7       $\mathcal{L}$.append($D_{\mathsf{r}}$)
8    **end**
9 **end**
   // Include different combinations of base functions:
10 **for** $b = 1$ **to** $b_{\mathsf{max}}$ **do**
11    **for** $d^2_{\mathsf{out}} = 0$ **to** $d^2_{\mathsf{max,out}}$ **do**
12       **for** $d^1_{\mathsf{out}} = 1$ **to** $d^1_{\mathsf{max,out}}$ **do**
13          **for** $d^2_{\mathsf{in}} = 0$ **to** $d^2_{\mathsf{max,in}}$ **do**
14             **for** $d^1_{\mathsf{in}} = 1$ **to** $d^1_{\mathsf{max,in}}$ **do**
15                **for** $B$ **as a list with** $b$ **elements of** $G_{\mathsf{max}}$ **do**
                  // Note that base functions can be contained in $B$ multiple times.
16                  $D$ = {'DegInputNumerator': $d^1_{\mathsf{in}}$, 'DegOutputNumerator': $d^1_{\mathsf{out}}$, 'DegInputDenominator': $d^2_{\mathsf{in}}$, 'DegOutputDenominator': $d^2_{\mathsf{out}}$, 'baseFunctions': $B$}
17                  $\mathcal{L}$.append($D$)
18             **end**
19          **end**
20       **end**
21    **end**
22    **end**
23 **end**
24 **return** $\mathcal{L}$

---

This strategy is comparable to the one proposed by Bartlett et al. (2023), called "Exhaustive Symbolic Regression". There, they iterate through a list of parameterized functions and use BFGS to identify the parameters. To create the list of parametrized functions, they construct every possible function using a given set of base operations and a predefined complexity. Notably, this results in more than 100,000 functions to evaluate for one-dimensional data, with the same set of base functions as we do, but without $\cos$. Our algorithm, however, only needs to search for the parameters of around 500 functions since it covers many at the same time by employing the global optimization strategy.

Due to this high complexity, Bartlett et al. (2023) state that they merely concentrate on one-dimensional problems and, thus, could benchmark their algorithm only on Feynman-I-6-2a ($y = \exp(\theta^2/2)/\sqrt{2pi}$), the only one-dimensional problem from the Feynman data set (Udrescu & Tegmark, 2020). This example shows the benefit of employing global search in the parameter space:

While ParFam needs five minutes of CPU time to compute the correct function, Bartlett et al. (2023) need 33 hours (150 hours, if the set of possible functions is not pre-generated).

## I HYPERPARAMETER SETTINGS SRBENCH GROUND-TRUTH PROBLEMS

The hyperparameter settings for the SRBench ground-truth problems are summarized in Table 7.

Table 7: The model and optimization parameters for the SRBench ground-truth problems for ParFam and DL-ParFam

| | | |
|---|---|---|
| **Model parameters** | Maximal Degree First Layer Numerator | 2 |
| | Maximal Degree First Layer Denominator | 2 |
| | Maximal Degree Second Layer Numerator | 4 |
| | Maximal Degree Second Layer Denominator | 3 |
| | Base functions | $\sqrt{}$, cos, exp |
| | Maximal potency of any variable (i.e., $x_1^4$ is excluded but $x_1^3 x_2$ is allowed) | 3 |
| **Optimization parameters** | Global optimizer | Basin-hopping |
| | Local optimizer | BFGS |
| | Maximal number of iterations global optimizer | 10 (1 for DL-ParFam) |
| | Maximal data set length | 1000 |
| | Regularization parameter $\lambda$ | 0.001 |

## J ADDITIONAL PLOTS FOR THE SRBENCH GROUND-TRUTH RESULTS

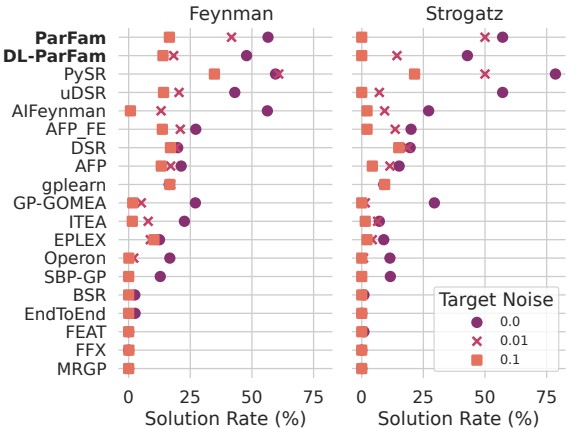

Figure 9: Symbolic solution rate on both SRBench ground-truth data sets separated.

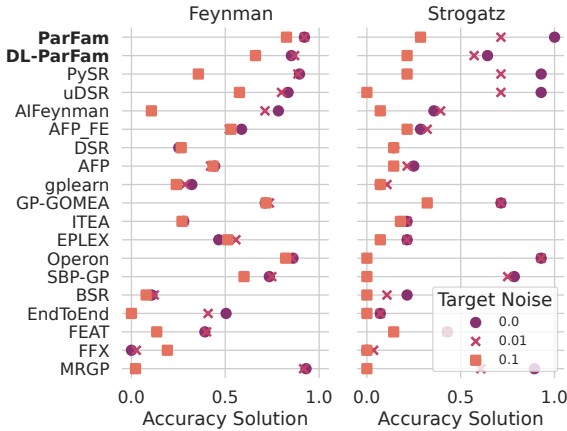

Figure 10: Accuracy solution rate (percentage of data sets with $R^2 > 0.999$ for the test set) on the SRBench ground-truth problems separately.

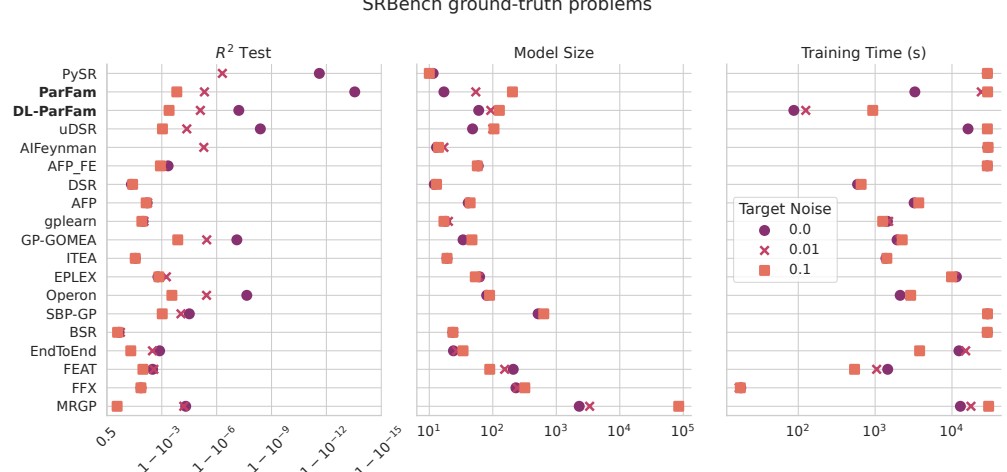

Figure 11: Results on the SRBench ground-truth problems. Points indicate the median test set performance on all problems. The $R^2$ Test for AIFeynman is missing on the plot since SRBench used a higher precision data type, such that AIFeynman achieved a median $R^2$ greater than $1 - 10^{-16}$.

## K    SENSITIVITY ANALYSIS FOR $\lambda$

In Table K, we present the results for ParFam on the ground-truth SRBench data sets for different values of $\lambda$. Note, that this has been done afterwards as a sensitivity analysis and not to choose $\lambda$. Our selection of $\lambda = 0.001$ was based on theoretical considerations and prior observations on toy examples while debugging ParFam. Table 8 shows that ParFam is robust with respect to $\lambda$. Surprisingly, the complexity of the learned formulas increases for increasing $\lambda$. This counterintuitive phenomenon might be due to various reasons. First, the 1 norm does not enforce sparsity but favors it, since it is only a proxy for it. So, a lower 1 norm does not necessarily imply a lower sparsity. Furthermore, the enumeration through the model parameters breaks the monotonous influence of the regularization. For example, a smaller parametric family might have been the best for a lower regularization parameter.

Table 8: Results of ParFam on the ground-truth SRBench data sets for different values of $\lambda$.

| $\lambda$ | Accuracy solution rate | Symbolic solution rate | Complexity |
|---|---|---|---|
| 0.0001 | 94.7% | 50% | 227 |
| 0.001 | 91.7% | 55.6% | 131 |
| 0.01 | 94.7% | 52.2% | 243 |

## L   BLACK-BOX DATA SETS: DL-PARFAM

The current implementation of DL-ParFam can only handle 9 independent variables at most. For this reason, we benchmark DL-ParFam against the other algorithms on the black-box data sets, which have at most 9 independent variables. The result on the remaining 50 data sets can be seen in Figure 12.

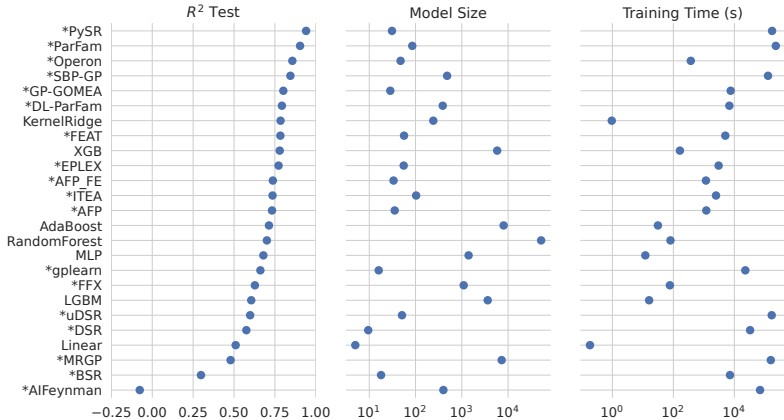

Figure 12: Median $R^2$, formula complexity, and training time on the 50 black-box problems from SRBench (La Cava et al., 2021) with at most 9 independent variables. The asterisk indicates that it is a symbolic regression method.

## M   COMPARING PARFAM TO EQL ON SRBENCH

As described in the introduction, EQL (Martius & Lampert, 2017) is the closest method to ParFam, since both make use of non-linear parametric models to translate SR to a continuous optimization problem. Because of this similarity, we believe that it is important to also show numerical comparisons between these two. Even though Sahoo et al. (2018) extended EQL to include the division operator, also their version of EQL (EQL with division) is not able to express the square root, logarithm, and exponential, which is why we created a reduced version of the ground-truth SRBench, which omits all equations using any of these base functions. In total, this covers 96 formulas. The results of EQL with division on these can be seen in Table 9.

To ensure a fair comparison for EQL, we first tried to run it using the default learning parameters and model parameter search recommended by the authors. However, since EQL will then quickly use up the computing budget given by SRBench (8 hours of CPU time) we tested EQL for multiple different hyperparameters on the first 20 problems from SRBench. We then chose the best-performing hyperparameters (epoch factor: 1000, penalty every: 50, maximal number of layers: 4, l1: 10) and reran the whole benchmark. This, together with the initial run using the recommended parameters, gives two formulas per equation. In the results shown in Table 9 we chose the formula with the better $R^2$ on the validation data set. Note, that we did not make use of the information, that the square root, logarithm, and exponential are never parts of the formulas when running ParFam, i.e.,

we included these base functions in the dictionary. In our experiments we used the code from the official EQL github repository (GNU General Public License v3.0).

Table 9: Results of ParFam and EQL with division (Martius & Lampert, 2017; Sahoo et al., 2018) on the 96 SRBench ground-truth equations, which do not include the square root, logarithm, and exponential.

|  | **Accuracy solution ($R^2 > 0.999$)** | **Symbolic solution** |
|---|---|---|
| ParFam | 93.8% | 69.8% |
| EQL | 75% | 16.7% |

## N    COMPARING DL-PARFAM TO NESYMRES ON SRBENCH

Since NeSymRes (Biggio et al., 2021) allows at most 3 independent variables, we compare it with DL-ParFam on a corresponding subset of SRBench. We perform the experiments with the same settings as the experiments shown in Section 3 and the same hyperparameters for DL-ParFam, which are reported in Appendix I. The hyperparameters for NeSymRes are reported in Appendix G. The results are summarized in Table 10, which shows that DL-ParFam outperforms NeSymRes in the symbolic solution rate as well as the accuracy solution rate while being more than 20 times faster, even though the model was trained for dimensions 1 to 9.

## O    COMPARISON WITH BAYESIAN OPTIMIZATION

The goal of DL-ParFam is to simplify the model-parameter search, which is usually done by performing a grid search for ParFam, where we start with testing simple configurations first and slowly increase the complexity of the parametric families. Another standard approach to accelerate the hyper-parameter optimization is Bayesian optimization (Shahriari et al., 2015). To evaluate the impact by pre-training a SET Transformer first, to guide the model-parameter selection, we compare the performance of DL-ParFam with ParFam with Bayesian optimization using Gaussian processes to guide the model-parameter selection and ParFam with structured grid search. The Bayesian optimization searches through the same model parameters as ParFam with grid-search (Algorithm 1), i.e., the values shown in Table 7. Table 11 presents the results on the ground-truth problems of SRBench. While Bayesian hyperparameter optimization manages to speed up the training as well, DL-ParFam outperforms it with respect to symbolic solution rate and training time.

## P    NGUYEN BENCHMARK

To compare ParFam with SPL (Sun et al., 2022) and NGGP (Mundhenk et al., 2021), which are the current state-of-the-art on some SR benchmarks (like Nguyen (Uy et al., 2011)), but no results of them on SRBench were reported, we evaluate ParFam on Nguyen. Interestingly, we observed that the original domain on which the data was sampled is not big enough to specify the functions, as

Table 10: Results on the subset of the SRBench ground-truth problems containing only expression with at most 3 variables. Following SRBench terminology, training time refers to the time each algorithm requires to compute a result for a specific problem, which corresponds to inference time for pre-trained methods.

|  | **Symbolic solution** | **Accuracy solution ($R^2 > 0.999$)** | **Training time** (in s) |
|---|---|---|---|
| DL-ParFam | 58.1% | 87.1% | 53 |
| NeSymRes | 48.7% | 59.0% | 1389 |

|  | Symbolic solution rate | Accuracy solution rate | Training time |
|---|---|---|---|
| Bayesian (max. 50 calls) | 34.9% | 85.3% | 7678s |
| Bayesian (max. 500 calls) | 38.0% | 89.1% | 10937s |
| DL-ParFam | 45.9% | 83.5% | 234s |
| Grid search | 55.6% | 93.2% | 12860s |

Table 11: Comparison of symbolic solution rate, accuracy solution rate, and training time on the ground-truth problems of SRBench for ParFam with model-parameter selection guided by Bayesian optimization, a SET Transformer, and grid search.

ParFam was able to find simple and near indistinguishable approximations to the data that are not the target formula. For example, it found $0.569x^2 - 0.742\sin(1.241x^2 - 2.059) - 1.655$ instead of $\sin(x^2)\cos(x) - 1$, since both are almost identical on the domain $[-1, 1]$. For this reason, we extended the data domain for some of the problems. The results for the Nguyen data set can be seen in Table 12. We used the hyperparameters shown in Table 13. Following Sun et al. (2022), from whom we take the results of the competitors, we use $\sin$ and $\exp$ as the standard basis functions for ParFam and add $\sqrt{}$ and $\log$ for the problems 7, 8, 11, and $8^c$. Note that the formula Nguyen-11 can not be expressed by ParFam and hence the symbolic solution rate is 0.

Table 12: Results on the Nguyen benchmarks. The results for ParFam are averaged over 6 independent runs. The results from SPL (Sun et al., 2022), NGGP (Mundhenk et al., 2021), and GP (a genetic programming-based SR algorithm) are taken from Sun et al. (2022).

| Benchmark | Expression | ParFam | SPL | NGGP | GP |
|---|---|---|---|---|---|
| Nguyen-1 | $x^3 + x^2 + x$ | 100% | 100% | 100% | 99% |
| Nguyen-2 | $x^4 + x^3 + x^2 + x$ | 100% | 100% | 100% | 90% |
| Nguyen-3 | $x^5 + x^4 + x^3 + x^2 + x$ | 100% | 100% | 100% | 34% |
| Nguyen-4 | $x^6 + x^5 + x^4 + x^3 + x^2 + x$ | 100% | 99% | 100% | 54% |
| Nguyen-5 | $\sin(x^2)\cos(x) - 1$ | 83% | 95% | 80% | 12% |
| Nguyen-6 | $\sin(x) + \sin(x^2 + x)$ | 83% | 100% | 100% | 11% |
| Nguyen-7 | $\log(x+1) + \log(x^2 + 1)$ | 100% | 100% | 100% | 17% |
| Nguyen-8 | $\sqrt{x}$ | 100% | 100% | 100% | 100% |
| Nguyen-9 | $\sin(x_0) + \sin(x_1^2)$ | 100% | 100% | 100% | 76% |
| Nguyen-10 | $2\sin(x_0)\cos(x_1)$ | 100% | 100% | 100% | 86% |
| Nguyen-11 | $x^y$ | 0% | 100% | 100% | 13% |
| Nguyen-12 | $x_0^4 - x_0^3 + 0.5x_1^2 - x_1$ | 100% | 28% | 4% | 0% |
| Nguyen-1$^c$ | $3.39x^3 + 2.12x^2 + 1.78x$ | 100% | 100% | 100% | 0% |
| Nguyen-2$^c$ | $0.48x^4 + 3.39x^3 + 2.12x^2 + 1.78$ | 100% | 94% | 100% | 0% |
| Nguyen-5$^c$ | $\sin(x^2)\cos(x) - 0.75$ | 83% | 95% | 98% | 1% |
| Nguyen-8$^c$ | $\sqrt{1.23x}$ | 100% | 100% | 100% | 56% |
| Nguyen-9$^c$ | $\sin(1.5x_0) + \sin(0.5x_1^2)$ | 100% | 96% | 90% | 0% |
| Average | | 91.2% | 94.5% | 92.4% | 38.2% |

Table 13: The model and optimization parameters for the Nguyen benchmark.

| | | |
|---|---|---|
| **Model parameters** | Maximal Degree Input Numerator | 2 |
| | Maximal Degree Input Denominator | 0 |
| | Maximal Degree Output Numerator | 6 |
| | Maximal Degree Input Denominator | 0 |
| | Base functions | cos, exp $(\sqrt{}, \log)$ |
| | Maximal potence of any variable | 6 |
| **Optimization parameters** | Global optimizer | Basin-hopping |
| | Local optimizer | BFGS |
| | Maximal number of iterations global optimizer | 30 |
| | Regularization parameter $\lambda$ | 0.1 |

Table 14: Results on SRSD-Feynman easy

|  | **Symbolic solution** | **Accuracy solution** ($R^2 > 0.999$) | **Training time** (in s) |
|---|---|---|---|
| ParFam | 76.7% | 93.3% | 7620 |
| DL-ParFam | 60% | 66.7% | 72 |
| PySR | 90% | 93.3% | 5796 |
| uDSR | 66.7% | 86.7% | 7470 |

Table 15: Results on SRSD-Feynman medium

|  | **Symbolic solution** | **Accuracy solution** ($R^2 > 0.999$) | **Training time** (in s) |
|---|---|---|---|
| ParFam | 47.5% | 80% | 8478 |
| DL-ParFam | 40% | 65% | 138 |
| PySR | 62.5% | 85% | 8022 |
| uDSR | 40% | 82.5% | 7740 |

# Q   SRSD FEYNMAN

Matsubara et al. (2024) introduced the SRSD-Feynman benchmarks (Creative Commons Attribution 4.0 International) building on the Feynman datasets (Udrescu & Tegmark, 2020) to substitute the artificial ranges and coefficients imposed by the original datasets with the physical ones, resulting in the same formulas with coefficients and variables ranging from $10^{-30}$ to $10^8$. The best-performing algorithms in their benchmark are PySR (Cranmer, 2023) and uDSR (Landajuela et al., 2022), which is why we test ParFam and DL-ParFam against those two.

We ran the experiments for the easy and medium difficulty for ParFam and DL-ParFam and the two best-performing algorithms from Matsubara et al. (2024) since these seem to be the strongest in the field currently, as also supported by our experiments on SRBench. We use the same settings as for the Feynman data sets (8CPUh and 1,000,000 function evaluations) and the same hyperparameters for all algorithms as shown in Appendix I and G.

The results are shown in Table 14 and 15. Note that our results differ from those reported in Matsubara et al. (2024) since we normalize the data for each algorithm. The results show that ParFam performs slightly worse than PySR but better than uDSR. Furthermore, DL-ParFam is only slightly worse than its competitors, while being up to 50 to 100 times faster. It is particularly interesting to see that DL-ParFam has a reasonable performance even on data sampled from a very different domain than the one it was trained on, probably due to the normalization of the input data during its training.

