# OpenReview forum: "ParFam -- (Neural Guided) Symbolic Regression via Continuous Global Optimization"
_ICLR.cc/2025/Conference — ICLR 2025 Poster_

### Official Review · Reviewer_tj4N · 2024-10-31

**Soundness:** 3
**Presentation:** 3
**Contribution:** 2
**Rating:** 5
**Confidence:** 3

**Summary:**

In this paper, the authors propose a novel architecture, ParFam, to solve symbolic regression through continuous optimization. The authors also introduce the use of a neural network to predict optimal hyperparameters for ParFam.

**Strengths:**

The strength of this paper is that the proposed ParFam shows high accuracy on the Feynman and Strogatz datasets from SRBench. The idea of using a neural network to predict optimal hyperparameters is interesting.

**Weaknesses:**

The weakness of ParFam is that the advantage of ParFam over EQL and some recent EQL variants is not clearly demonstrated. The authors claim that EQL uses linear layers, while ParFam uses rational layers. However, EQL is not limited to using unary functions in basis functions, and if a division function were included in EQL, it would closely resemble ParFam.

**Questions:**

Here are some questions that need to be addressed:
1. The idea of using a division operator in EQL was proposed in ICML 2018 [1]. Please provide a comparison between ParFam and EQL with division operators to ensure a fair comparison.
2. Recent works have incorporated neural architecture search with EQL [2] [3]. These works appear to be a superset of the proposed method. The architecture proposed here is manually designed within the neural architecture search space. It is unclear what advantage ParFam has over these variants of EQL.
3. The idea of predicting hyperparameters using a transformer is interesting. In ParFam, the default method for hyperparameter optimization is grid search. However, Bayesian optimization is a more common approach. Please provide a comparison of the speedup achieved by hyperparameter optimization using a pre-trained transformer versus Bayesian optimization.
4. The training time of ParFam on the black-box datasets from SRBench is not shown. Please provide this information.
5. For the limitation related to high-dimensional data, it is claimed that "the number of parameters grows exponentially with the number of variables." However, from Figure 1, it is unclear why the number of parameters would grow exponentially with the number of variables. Please clarify.

References:

[1]. Sahoo, Subham, Christoph Lampert, and Georg Martius. "Learning equations for extrapolation and control." International Conference on Machine Learning. PMLR, 2018.

[2]. Dong, Junlan, et al. "Evolving Equation Learner for Symbolic Regression." IEEE Transactions on Evolutionary Computation (2024).

[3]. Li, Wenqiang, et al. "A Neural-Guided Dynamic Symbolic Network for Exploring Mathematical Expressions from Data." Forty-first International Conference on Machine Learning.

---

> ### Author Response · Authors · 2024-11-21
>
> [1/3]
>
> We thank Reviewer tj4N for their time and thoughtful comments on our paper and a chance to address their concerns. We uploaded a revised version of the paper to follow the ideas and recommendations of the reviewers.
>
> __EQL is not limited to using unary functions in basis functions, and if a division function were included in EQL, it would closely resemble ParFam. The idea of using a division operator in EQL was proposed in ICML 2018 [1]. Please provide a comparison between ParFam and EQL with division operators to ensure a fair comparison.__
>
> We appreciate this observation and apologize for any confusion regarding our comparison. We cite both the original EQL paper [4] and the ICML 2018 paper [1], which extends EQL to include the division operator, in our introduction to emphasize that we are referring to EQL with the division operator. We hope that this is clearer by our revision in the introduction.
>
> The primary distinction between ParFam and EQL (with division) lies in how multiplication and division operations are implemented:
>
> - **ParFam**: Incorporates multiplication and division directly inbetween layers through high-order rational functions, enabling seamless modeling of products and powers involving multiple variables (both in the numerator and denominator).
> - **EQL**: Introduces multiplication and division as activation functions that take only two inputs at a time, necessitating multiple layers to represent more complex products, quotients, and powers.
>
> This architectural difference simplifies the optimization in ParFam by reducing the number of layers, which can enhance interpretability, stability, and efficiency. For example, the additional layers in EQL may lead to exploding gradients when handling exponential functions. Note that for this reason, EQL is currently not able to handle the exponential, a limitation not shared with ParFam.
>
> We also want to clarify that our experiments in Appendix M (Appendix L in the revised version) already include EQL with the division operator. We also explicitly state this in revised version of the paper to avoid confusion.
>
> __Recent works have incorporated neural architecture search with EQL [2]  [3]. These works appear to be a superset of the proposed method. The architecture proposed here is manually designed within the neural architecture search space. It is unclear what advantage ParFam has over these variants of EQL.__
>
> Thank you for pointing us to these relevant works [2, 3], which we included in the revised introduction. While both works introduce neural architecture search methods for EQL, they remain distinct from ParFam.
>
> Dynamic Symbolic Network employs a recurrent neural network to dynamically select EQL architectures. However, the resulting architectures still conform to EQL's structure, with multiple hidden layers, which contrasts with ParFam's compact structure integrating multiplication and division directly in the layers.
>
> EQL with Evolved Basis Functions [2] uses genetic programming to evolve additional basis functions for EQL. While this approach extends EQL's capabilities, it retains EQL's linear connections between activation functions, necessitating multiple hidden layers for complex expressions. It would be interesting to include additional experiments to benchmark this algorithm against ParFam, however, there is no source code public.

---

> > ### Author Response · Authors · 2024-11-21
> >
> > [2/3]
> >
> > __The idea of predicting hyperparameters using a transformer is interesting. In ParFam, the default method for hyperparameter optimization is grid search. However, Bayesian optimization is a more common approach. Please provide a comparison of the speedup achieved by hyperparameter optimization using a pre-trained transformer versus Bayesian optimization.__
> >
> > This is an excellent suggestion. We investigated Bayesian hyperparameter optimization using Gaussian progresses implemented by skopt (skopt.gp_minimize) and ran the experiments on the ground-truth problems with. The results are shown in the table below.
> >
> > |                           | Symbolic solution rate | Accuracy solution rate | Training time |
> > | ------------------------- | ---------------------- | ---------------------- | ------------- |
> > | Bayesian (max. 50 calls)  | 34.9%                  | 85.3%                  | 7678s         |
> > | Bayesian (max. 500 calls) | 38.0%                  | 89.1%                  | 10937s        |
> > | DL-ParFam                 | 45.9%                  | 83.5%                  | 234s          |
> > | Grid search               | 55.6%                  | 93.2%                  | 12860s        |
> > While Bayesian hyperparameter optimization manages to speed up the training, DL-ParFam outperforms it with respect to symbolic solution rate and training time. The strong drop in performance for Bayesian hyperparameter optimization in comparison with grid search surprised us and we hypothesize that it stems from the fact that the grid search was very deliberately chosen to start with small parameteric families and to slow grow them over time, as some of the big parametric families are hard to optimize and use up a lot of training time.
> >
> > As we think that it is an interesting alternative for hyper-parameter optimization, we include it in as Appendix N of the revised paper.
> >
> >
> > __The training time of ParFam on the black-box datasets from SRBench is not shown. Please provide this information.__
> >
> > We apologize for omitting this information. We did not define any early stopping condition for the black-box experiments, such that all algorithms we tested on the blackbox datasets (ParFam, pysr, and uDSR) used the whole budget (24CPUh), which is why we didn't deem the training time to be meaningful.
> >
> > However, we understand the importance of transparency and included these training times in the revised manuscript for completeness, see Figure 5.
> >
> > __For the limitation related to high-dimensional data, it is claimed that "the number of parameters grows exponentially with the number of variables." However, from Figure 1, it is unclear why the number of parameters would grow exponentially with the number of variables. Please clarify.__
> >
> > Thank you for raising this important point! The claim that the number of parameters grows exponentially with the number of variables was indeed imprecise. Below, we clarify the actual relationship:
> >
> > Since ParFam's parameters are the coefficients of the polynomials of the rational functions, we have to compute the number of coefficients of a polynomial in $n$ variables with degree $d$:
> >
> > $$p(x)=\sum_{\alpha\in\mathbb{N}^n: |\alpha|\leq d} a_\alpha x^\alpha$$
> > The number of coefficients corresponds to the number of multi-indices $\alpha\in\mathbb{N}^n$ satisfying $|\alpha|=\sum_{i=1}^n\alpha_i\leq d$. Using combinatorics (how many possibilities are there to choose $d$ elements from a set of $n$ elements with repitition), we get that there are
> > $$\sum_{k=0}^d\binom{n-1+k}{k}=\binom{n+d}{d}=\frac{(n+d)!}{(d)!(n)!}$$
> > coefficients of $p$. For $n>>d$, the growth rate is approximately $n^d$, showing that the number of parameters grows polynomially, not exponentially, with the number of variables $n$.
> >
> > Our earlier statement incorrectly assumed an exponential growth due to the factorial terms in the binomial coefficient. We revised this in the manuscript by changing the previous sentence in the following way:
> >
> > "Another limitation of ParFam is that solving high-dimensional problems ($>$$10$ independent variables) with a global optimizer is computationally expensive, as the number of parameters grows in $O(n^d)$, where $n$ is the number of variables and $d$ is the maximal degree of the polynomials involved."
> >
> > Thank you for pointing out this error!

---

> > > ### Author Response · Authors · 2024-11-21
> > >
> > > [3/3]
> > >
> > > [1] Subham Sahoo, Christoph Lampert, and Georg Martius. Learning equations for extrapolation and control. In Proceedings of the 35th International Conference on Machine Learning, volume 80 of Proceedings of Machine Learning Research, pp. 4439–4447. PMLR, 2018.
> > >
> > > [2]. Dong, Junlan, et al. "Evolving Equation Learner for Symbolic Regression." IEEE Transactions on Evolutionary Computation (2024).
> > >
> > > [3]. Li, Wenqiang, et al. "A Neural-Guided Dynamic Symbolic Network for Exploring Mathematical Expressions from Data." Forty-first International Conference on Machine Learning.
> > >
> > > [4] Georg Martius and Christoph H. Lampert. Extrapolation and learning equations. In 5th International Conference on Learning Representations, Workshop Track Proceedings. 2017

---

> > ### Comment · Reviewer_tj4N · 2024-11-22
> >
> > Thanks for the response. I still have three questions related to questions 2, 3, and 4:
> >
> > **Comparison with DySymNet:**
> > The gradient explosion issue in EQL is indeed a limitation. From my understanding, ParFam uses complex activation functions, specifically rational functions, allowing even two layers to outperform EQL and avoid gradient explosion. This is a valid claim.
> >
> > However, DySymNet is still worth considering as a strong baseline. According to the results in the DySymNet paper, DySymNet outperforms EQL by a significant margin [1]. Since DySymNet is a neural architecture search algorithm, it could potentially circumvent the gradient explosion issue by avoiding designs that lead to such problems. Additionally, it might be capable of automatically discovering ParFam-like architectures.
> >
> > The source code for DySymNet is available at https://github.com/AILWQ/DySymNet. Please consider comparing ParFam with DySymNet.
> >
> > **Bayesian Optimization:**
> > Based on your new response, it seems that the grid search was deliberately designed to start with small parametric families and gradually expand over time, as some larger parametric families are harder to optimize and require significant training time. However, this is unclear. I couldn’t find the parameter grid in the supplementary material. Specifically, Table 7 does not appear to be a grid.
> >
> > Additionally, regarding the comparison with Bayesian optimization, what search space was used for Bayesian optimization? Is it consistent with the grid search space? From the results, increasing Bayesian optimization to 500 maximum iterations seems to only slightly increase the runtime compared to Bayesian optimization with 50 iterations. What is the reason for this?
> >
> > By the way, the newly added Table 11 has a style that is inconsistent with the other tables. Please revise this for uniformity.
> >
> > **Training Time on the Black-Box Dataset:**
> > Based on the new results, it appears that ParFam does not have an advantage in terms of R2 score, model size, or training time. Please consider reporting the results of DL-ParFam on the black-box SRBench dataset.
> >
> > [1]. Li, Wenqiang, et al. "A Neural-Guided Dynamic Symbolic Network for Exploring Mathematical Expressions from Data." Forty-first International Conference on Machine Learning.

---

> ### Author Response · Authors · 2024-11-25
>
> Thank you for your thoughtful questions and patience.
>
> __DySymNet__
>
> We tested DySymNet on the SRBench ground-truth datasets (without noise) using the hyperparameters specified in Table 4 of the DySymNet paper. However, including 1 in the list for `"Number library of layers"` and `"Number library of operators for each layer"`, as it was done in the paper, led to the following error:
>
> > _AssertionError: Error: the input dim of the first step is not equal to the max dim._
>
> As a workaround, we used the default parameters from the official DySymNet repository ([link](https://github.com/AILWQ/DySymNet)):
>
> - Layers: `[2, 3, 4, 5]`
> - Operators per layer: `[2, 3, 4, 5, 6]`
>
> Unfortunately, this configuration produced unsatisfactory results:
>
> - Symbolic solution rate: **3.9%**
> - Accuracy solution rate: **5.5%**
>
> We resolved the error by setting `"input_size"` in the config file to 1. However, this parameter is undocumented in the paper and online resources, so the correct value remains unclear. We’re rerunning the experiments with this adjustment and will share results soon. Any suggestions for improving DySymNet’s performance or correctly addressing this issue are welcome.
>
> __Bayesian optimization:__
> - **Grid search:** Table 7 only shows the *maximal degrees* that can be chosen *during the model-parameter search*. On page 9, lines 468-470, we state the following:
>
>     > “This choice [of model parameters] results in a parametric family with hundreds of parameters, making it challenging for global optimization. To address this issue, we iterate for ParFam through various smaller parametric families, each contained in this larger family (details in Appendix H).”
>
> This iteration process (grid search) is detailed in Algorithm 1. We apologize for any confusion about this topic.
>
> - **Bayesian optimization search space:** The search space for Bayesian optimization is identical to grid search. We clarified this in Appendix N:
>
>     > “The Bayesian optimization searches through the same model parameters as ParFam with grid search (Algorithm 1), i.e., the values shown in Table 7.”
>
> - **Training Time:** Using 500 calls does not increase training time tenfold due to:
>
>     1. Early Stopping: The process stops early if a simple, accurate formula is found.
>     2. Time Constraints: Each run is capped at 8 hours (28,800 seconds), often preventing all 500 calls.
>
> Thank you for pointing us to the inconsistency in the table formatting. We changed this in the revised version.
>
> **Black-Box Dataset:**
>
> We want to emphasize that it was not our goal to introduce ParFam as a method that outperforms all competitors and we do not claim this in our work.
>
> However, it is worth noting that PySR and ParFam are the only methods consistently among the top five performers on both black-box and ground-truth datasets. Other methods exhibit significant performance variability between datasets. The consistent performance of PySR and ParFam shows their robustness and reliability across diverse tasks. Furthermore, it is important to note that PySR is an extremely optimized algorithm [1]: The algorithm itself is rooted in decades-long research in Genetic Programming, the most mature area in SR, and the implementation has been highly optimized in Julia. In contrast, ParFam builds upon a relatively new field and is implemented in Python/PyTorch with a focus on conceptual clarity rather than hyper-optimization. Despite these differences, ParFam’s competitive results highlight its potential and the promise of continuous optimization methods in SR.
>
> DL-ParFam's architecture can only take in 9 dimensional problems as input (see Appendix D). For this reason we filtered the black-box experiements to remove any data-set with more than 9 features. The results can be seen in Figure 12 in Appendix L in the newly revised version.
>
> [1] Cranmer, Miles. "Interpretable machine learning for science with PySR and SymbolicRegression.jl." _arXiv preprint arXiv:2305.01582_ (2023).

---

> > ### Author Response · Authors · 2024-11-26
> >
> > We thank the reviewer again for the fruitful discussion. We now finished our experiments with DySymNet.
> >
> > Unfortunately, our experiment using the exact hyperparameters reported in the DySymNet paper (with the fix of `input_size=1`) did not yield better results. The paper does not specify any pre-processing steps or the dataset size, so we tested with and without normalization and across different dataset sizes (500, 1,000, and 10,000 samples). However, we were unable to achieve an accuracy above 6% or a symbolic solution rate above 4%. Additionally, no instructions are provided in the repository for reproducing DySymNet's results on the SRBench datasets.
> >
> > If the reviewer has any insights or suggestions for improving performance, we would be happy to implement them.

---

> > > ### Comment · Reviewer_tj4N · 2024-11-30
> > >
> > > Thanks to the authors for your efforts. From Figure 1 of the DySymNet paper, it appears that DySymNet demonstrates better performance than Operon in terms of both model size and $R^2$ score on black-box datasets. However, the proposed method performs worse than Operon on these datasets. This raises questions about the advantage of using ParFam instead of a neural architecture search-based symbolic regression method like DySymNet. Based on the current evidence, I must maintain my current score.
> > >
> > > Regarding the unsatisfactory results on DySymNet, you may consider verifying whether DySymNet has been used correctly by examining the training error.

---

> > > > ### Author Response · Authors · 2024-11-30
> > > >
> > > > We thank Reviewer tj4N for the extended discussion!
> > > >
> > > > We aimed to identify the reasons for the disparity between the performance of DySymNet reported in the paper and in our experiments. As suggested by the reviewer, we checked the training error reported directly by the DySymNet package and it fits the training and test error computed by us afterwards, indicating that the data is transferred correctly to DySymNet and we evaluate it correctly afterwards. We identified 2 main points we believe to cause the difference:
> > > > - **We believe that DySymNet does not follow the time and evaluation limits as defined by the SRBench paper**: The DySymNet paper does not report any training time nor does it make any statements on training time limits or the number of function evaluation limits, even though these budgets are essential to the SRBench benchmark. Instead, DySymNet reports the hyperparameters used in Table 4. These state that they use 10,000 training epochs for the SymNet during the first stage ($n_1$) and 10,000 training epochs during the second stage ($n_2$). Unfortunately, the paper does not report the training epochs and batch size for the RNN, however, the GitHub states the default parameters of DySymNet, which mostly agrees with Table 4 in the paper. This list of default hyperparameters states that one should use DySymNet with 500 "epochs for sampling" and 10 as the "Size for a batch sampling". This fits to Figure 6 in the DySymNet paper, which shows the training curve of the RNN and displays it over 500 epochs. It also shows that the performance reaches convergence only after ~250 epochs. In our experiments with DySymNet we added a time limit (a feature that is not implemented in the github) and stop the experiments after the standard time limit of 8CPUh set in SRBench. This resulted in stopping most runs after a few epochs (<10 epochs) which is not enough for the RNN to converge in general as indicated in Figure 6. Furthermore, the SRBench paper specifies a budget for the number of function evaluations of 1,000,000, since time limit is a hardware and implementation-dependent measure. If the DySymNet was trained using 10,000 epochs in each stage of the SymNet, with sampling batches for the controller RNN of size 10 (this means that 10 SymNet are sampled and trained during each epoch of the controller), and with 500 sampling epochs of the controller RNN, then DySymNet needs at least $2\cdot10,000\cdot10\cdot500=100,000,000$ evaluations which far exceeds the budget given by SRBench. We did not enforce this limit on DySymNet in our experiments though, but with the reduced number of epochs, the reported results are in a reasonable range.
> > > > - **DySymNet has a hard-coded early stopping criterion at $R^2>0.99$**: Even though we believe that the change in performance is mostly due to the enforced training budget, the reported accuracy is still surprisingly low. This might be caused by a hard-coded early stopping criterion at $R^2>0.99$, which prevents most runs to find a formula with higher accuracy or continue to try to find the symbolic solution. Note that this fits to the DySymNet paper, since they report the accuracy solution for $R^2>0.99$ instead of $R^2>0.999$ as done in the SRBench paper and omit the symbolic solution rate completely. To make it a fair comparison to the other algorithms benchmarked, we removed the early stopping criterion and are currently rerunning the experiments. However, we do not expect the accuracy solution for $R^2>0.999$ to increase further than ~53%, since this was the ratio of problems for which DySymNet managed to find a formula with $R^2>0.99$ and stoped early. For a comparison, ParFam reached for 93% of the function $R^2>0.999$ and for 99% it reached $R^2>0.99$.
> > > >
> > > > In conclusion, we did our best to enable a fair comparison between ParFam and DySymNet, using the official GitHub of DySymNet and the hyperparameters when possible as defined in the paper and if not as suggested in the GitHub. We checked that the data is correctly used by DySymNet and we use the computed formula afterwards correctly. We implemented a time limit for DySymNet since this is a crucial step in SR to make the results comparable. We performed the experiments on the same hardware as for ParFam, such that the time limit should be perfectly comparable. All of the runs are stopped before using all the epochs as specified by the paper/github either due to the time limit or the early stopping. Even without the early stopping, DySymNet won't outperform ParFam on this dataset since only ~53% were stopped early. The most important difference seems the possible lack of time limits enforced in the paper. To reach their number of epochs we would need a 50-100 times higher time limit and 100 times higher evaluation limit which does not make for a fair comparison.

---

> > > > > ### Author Response · Authors · 2024-12-02
> > > > >
> > > > > We thank the reviewer again for the interesting discussion.
> > > > >
> > > > > As described before, we reran experiments on the SRBench ground-truth problems with DySymNet without the hard-coded early stopping. As expected the accuracy solution (for $R²>0.999$) increased slightly to 15%, but is still far below other algorithms, as ParFam, for example, reaches 93%. The symbolic solution rate remained below 5%.
> > > > >
> > > > > Therefore, we strongly believe that the difference in performance comes from using the adequate time limits as specified in the SRBench paper and, therefore, see an advantage of the architecture defined in ParFam over neural architecture search-based symbolic regression methods like DySymNet.

---

### Official Review · Reviewer_3uKW · 2024-11-02

**Soundness:** 2
**Presentation:** 3
**Contribution:** 2
**Rating:** 6
**Confidence:** 3

**Summary:**

This paper presents ParFam, a novel approach to symbolic regression that reformulates the discrete optimization problem into a continuous one using parametric families of functions. The DL-ParFam extension introduces a neural-guided approach that incorporates a pre-trained Set Transformer to accelerate the optimization process. The method is evaluated on standard symbolic regression benchmarks and shows competitive performance.

**Strengths:**

- The authors provide a thorough theoretical analysis of ParFam's expressivity, demonstrating that it can still represent a substantial proportion of functions of a given complexity.
- Results on standard benchmark datasets demonstrate ParFam's strong performance in terms of symbolic recovery rate and accuracy.

**Weaknesses:**

- The pre-training phase for DL-ParFam is computationally expensive, though it only needs to be done once.
- In the figure 4, the label "Symbolic Solution Rate" may be incorrect and should be "Symbolic Recovery Rate".
- As shown in Figure 4, symbolic solution rate drops significantly when noise is introduced. This suggests that the methods are sensitive to even small levels of noise, which could limit their robustness in real-world scenarios where data is often noisy.
- The method still faces challenges when dealing with high-dimensional problems (more than 10 variables). As the number of parameters increases exponentially with the number of variables, optimization becomes more computationally intensive.
- The performance of the method is highly dependent on model parameter choices, such as the degree of polynomials and basis functions, requiring some prior knowledge or extensive experimentation to determine the optimal parameters.

**Questions:**

- For DL-ParFam, how robust is the pre-trained model to out-of-distribution data? Are there certain types of functions where it consistently fails to provide useful guidance?
- Could the authors elaborate on how ParFam's performance scales with the number of variables?

---

> ### Author Response · Authors · 2024-11-21
>
> We thank Reviewer JBKY for their detailed review and thoughtful feedback. We uploaded a revised version of the paper to follow the ideas and recommendations of the reviewers.
>
> __In the figure 4, the label "Symbolic Solution Rate" may be incorrect and should be "Symbolic Recovery Rate".__
>
> Thank you for pointing us to this inaccuracy. We indeed defined the metric as symbolic recovery rate in the text and symbolic solution rate in the image. We changed the text to symbolic solution rate to follow the terminology introduced by SRBench.
>
> __The performance of the method is highly dependent on model parameter choices, such as the degree of polynomials and basis functions, requiring some prior knowledge or extensive experimentation to determine the optimal parameters.__
>
> We agree that the performance of ParFam depends on model parameter choices such as the degree of polynomials and the choice of basis functions. This reliance on parameter selection is an inherent limitation of structural approaches to symbolic regression. However, our experiments demonstrate that iterating through potential model parameters can be done efficiently within a reasonable time frame and does not take longer than other approaches, e.g., based on GP.
>
> Furthermore, we introduced DL-ParFam to automate and speed up this process.
>
> __For DL-ParFam, how robust is the pre-trained model to out-of-distribution data?__
>
> This is an excellent question and highly relevant to any pre-training-based method. The Feynman dataset itself is OOD data for DL-ParFam as it was trained on synthetic data, which shows that DL-ParFam performs also well on OOD data.
>
> To directly investigate the robustness of the SET Transformer in DL-ParFam, which is the pre-trained part of DL-ParFam, we conducted additional analyses to measure its performance. Specifically, we evaluated how often the SET Transformer correctly predicts model parameters for synthetic training datasets and the Feynman dataset, considering the top $k$ most likely predictions. The results are shown below:
>
> |           | Top 1 | Top 3 | Top 5 | Top10 |
> | - | - | - | - | - |
> | Synthetic | 31.4% | 50.2% | 61.8% | 71.2%  |
> | Feynman   | 30.4% | 38.0% | 40.5% | 45.6%  |
> The results indicate that while the SET Transformer used for DL-ParFam generalizes well to OOD data for its top predictions, there is room to optimize the synthetic training data further to improve its generalization. Note that "correct model parameters" refer to those spanning the parametric family with the minimal number of parameters covering the target function. Thus, DL-ParFam can sometimes recover the correct function without using the exact "correct" model parameters. We added this analysis in Appendix D.
>
> __Are there certain types of functions where DL-ParFam consistently fails to provide useful guidance?__
>
> One consistent challenge for DL-ParFam on the Feynman dataset is handling functions that include square roots, especially in expressions like $\sqrt{x_1^2+x_2^2}$. We hypothesize that this arises from the interplay between square and square-root operators, which is prevalent in the Feynman dataset.
>
> __Could the authors elaborate on how ParFam's performance scales with the number of variables?__
>
> We thank the reviewer for this insightful question. To evaluate how ParFam's performance scales with the number of variables, we split the ground-truth experiments by dimensionality and compared ParFam against PySR as a baseline. Below are the symbolic solution rates and accuracy solution rates for each algorithm (Note that there is only one dataset with 1, 8, and 9 dimensions and, therefore, these values are not that reliable.):
>
> **Symbolic Solution Rate**
>
> | Algorithm\Dimension | 1    | 2   | 3   | 4   | 5   | 6   | 7   | 8    | 9    |
> | ------------------- | ---- | --- | --- | --- | --- | --- | --- | ---- | ---- |
> | PySR                | 100% | 79% | 68% | 53% | 50% | 33% | 0%  | 100% | 0%   |
> | ParFam              | 100% | 72% | 70% | 53% | 35% | 11% | 0%  | 0%   | 100% |
>
>
> **Accuracy Solution Rate**
>
> | Algorithm\Dimension | 1    | 2    | 3    | 4   | 5   | 6   | 7   | 8    | 9    |
> | ------------------- | ---- | ---- | ---- | --- | --- | --- | --- | ---- | ---- |
> | PySR                | 100% | 97%  | 100% | 94% | 88% | 67% | 17% | 100% | 0%   |
> | ParFam              | 100% | 100% | 97%  | 97% | 85% | 78% | 33% | 100% | 100% |
>
>
> These results show a notable decline in both metrics for both methods for high-dimensional data, which is expected given the increased complexity. Interestingly, PySR experiences a more significant decline in accuracy solution rate, while ParFam's symbolic solution rate suffers more.

---

### Official Review · Reviewer_ZXN3 · 2024-11-02

**Soundness:** 3
**Presentation:** 2
**Contribution:** 2
**Rating:** 3
**Confidence:** 4

**Summary:**

In this paper, we propose a symbolic regression algorithm ParFam, which treats symbolic regression problem as a global optimization problem. Compared with the traditional methods that treat symbolic regression problem as a combinatorial optimization problem, ParFAM improves the search efficiency and has the potential to solve high-dimensional problems

**Strengths:**

This paper proposes DL-ParFam, which uses a pre-trained model to predict ParFam's parameters, effectively improving its training efficiency. The authors claim that the training speed is 100 times faster. But the thinking of DL - ParFam and this article is a bit like (https://doi.org/10.48550/arXiv.2309.13705),

**Weaknesses:**

##### Weaknesses

1. Mentioned in the article with global optimization to solve the problem of symbolic regression is more efficient, but this is not the first to use global optimization algorithm to solve the problem of symbolic regression algorithm, for example, EQL (https://doi.org/10.1109/TNNLS.2020.3017010), MetaSymNet (https://doi.org/10.48550/arXiv.2311.07326). And this paper does not compare with these two algorithms. Please analyze the advantages of ParFam over the above two methods.

2. At the beginning of the **introduction**, the paper mentioned that the simplicity of the expression is very important, but the paper did not evaluate the expression complexity of the algorithm.
3. Why did the author only add the noise level of 0.01 in the anti-noise experiment? However, in many other symbolic regression algorithms, the anti-noise experiment is more adequate. I think the noise level should be increased to the order of 0.1 to better test the anti-noise ability of ParFam.

**Questions:**

##### Questions

1. It is not clear to me how the training data for the DL-ParFam is collected. Please describe it in more detail in the article and provide more details.
2. To reorganize what is the main innovation of this paper, I don't think using the global optimization problem as a knot symbolic regression problem is an innovation of this paper.
3. The article says that its ability to deal with high-dimensional symbolic regression is stronger than the existing algorithms, so does the article test and compare the ability of each algorithm to deal with high-dimensional symbolic regression problems?
4. Why does DL-ParFam not compare inference time with pre-trained symbolic regression methods represented by **Neural Symbolic Regression that Scales** and **End-to-end Symbolic Regression with Transformers**?

---

> ### Author Response · Authors · 2024-11-21
>
> [1/2]
>
> We thank Reviewer ZXN3 for their time and thoughtful comments on our paper. We greatly appreciate the opportunity to address these concerns and improve our manuscript. We uploaded a revised version of the paper to follow the ideas and recommendations of the reviewers.
>
> __The thinking of DL - ParFam and this article is a bit like (A Neural-Guided Dynamic Symbolic Network for Exploring Mathematical Expressions from Data)__
>
> Thank you for bringing up this interesting reference. Both DL-ParFam and DySymNet rely on neural networks to guide the architecture search for symbolic regression. We include DySymNet in the related work section to highlight these connections. However, note, that the methods differ fundamentally in how they approach architecture prediction. While DL-ParFam employs pre-training to predict the architecture, DySymNet follows the idea of DSR [1] and uses an RNN recursively and retrains it for each instance using RL.
>
> __Mentioned in the article with global optimization to solve the problem of symbolic regression is more efficient, but this is not the first to use global optimization algorithm to solve the problem of symbolic regression algorithm, for example, EQL, MetaSymNet. And this paper does not compare with these two algorithms. Please analyze the advantages of ParFam over the above two methods.__
>
> First we want to emphasize that we compare with EQL in the last paragraph of the introduction and compare empirically with EQL in Appendix M (Appendix L in the revised version), which shows that ParFam strongly outperforms EQL. We follow the reviewers suggestion and include a discussion of MetaSymNet in the revised version.
>
> Therein, we acknowledge that other approaches, such as EQL and MetaSymNet, also use continuous optimization techniques for symbolic regression. __However, the formulation of our continuous optimization framework differs significantly from EQL and MetaSymNet.__ It is important to note that using this contribution leads to state of the art results. We revised the contributions section to ensure clarity, emphasizing that our focus is on introducing a novel way to translate the symbolic regression problem into a continuous optimization framework and showing its advantages over prior methods by adding following sentence:
>
> "While ParFam is not the first method to employ continuous optimization for symbolic regression, it aims to enhance the translation to the continuous space. By doing so, ParFam becomes the first SR method based on continuous optimization to achieve state-of-the-art performance."
>
> Following the recommendation by Reviewer JBK, we also include a more detailed description of the motivation and advantages of the ParFam's architecture in the last paragraph of Section 2.1.1:
>
> "the main motivation for the proposed architecture is that it allows ParFam to work with a single hidden layer, due to the high approximation qualities of rational functions and the general structure of common physical laws. Employing a single hidden layer offers several advantages: it reduces the number of parameters, simplifies optimization by mitigating issues such as exploding or vanishing gradients caused by nested functions, and enhances interpretability since it avoids complicated and uncommon compositions such as $\sin\circ\cos$, which many algorithms enforce to avoid as well."
>
> __At the beginning of the introduction, the paper mentioned that the simplicity of the expression is very important, but the paper did not evaluate the expression complexity of the algorithm.__
>
> We provide explicit complexity measures (termed "Model Size" following SRBench) in Figure 10 in Appendix K (Figure 11 in Appendix J in the revised version) for ground-truth problems and in Figure 11 in Appendix O (Figure 5 in Section 3 in the revised version) for black-box problems. Additionally, in line with SRBench, we report the Symbolic Solution Rate as a primary metric for the ground-truth problems, which serves as a proxy for expression complexity since symbolic solutions are inherently compact.
>
> __Why did the author only add the noise level of 0.01 in the anti-noise experiment? However, in many other symbolic regression algorithms, the anti-noise experiment is more adequate. I think the noise level should be increased to the order of 0.1 to better test the anti-noise ability of ParFam.__
>
> We acknowledge the importance of testing against higher noise levels. In our initial experiments, we limited ourselves to one noise setting ($0.01$) due to the computational expense of these tests. This choice was informed by SRBench, which used three noise levels ($0.001$, $0.01$, and $0.1$). To address this concern, we are currently running experiments with a higher noise level ($0.1$) to further evaluate ParFam's robustness.

---

> > ### Author Response · Authors · 2024-11-21
> >
> > [2/2]
> >
> > __It is not clear to me how the training data for the DL-ParFam is collected. Please describe it in more detail in the article and provide more details.__
> >
> > We expanded this explanation in Appendix C to provide a more detailed description in the revised manuscript. We hope that this clarifies the confusion.
> >
> > __To reorganize what is the main innovation of this paper, I don't think using the global optimization problem as a knot symbolic regression problem is an innovation of this paper.__
> >
> > We agree that global optimization for symbolic regression is not new, which we also do not claim in our paper. Our innovation lies in improving the translation of symbolic regression into a continuous optimization problem, making it a competitive alternative to genetic programming-based methods. Our extensive experiments demonstrate that ParFam achieves state-of-the-art performance on benchmark datasets and significantly outperforms previous approaches like EQL.
> >
> > __The article says that its ability to deal with high-dimensional symbolic regression is stronger than the existing algorithms, so does the article test and compare the ability of each algorithm to deal with high-dimensional symbolic regression problems?__
> >
> > We did not intend to imply that ParFam outperforms existing algorithms in high-dimensional symbolic regression. On the contrary, we explicitly state in the limitations section:
> >
> > _"Another limitation of ParFam is that solving high-dimensional problems (>10 independent variables) with a global optimizer is computationally expensive, as the number of parameters grows exponentially with the number of variables."_
> >
> > If the reviewer identifies any instance where the manuscript suggests otherwise, we will correct it promptly.
> >
> > __Why does DL-ParFam not compare inference time with pre-trained symbolic regression methods represented by Neural Symbolic Regression that Scales and End-to-end Symbolic Regression with Transformers?__
> >
> > Following SRBench terminology, "training time" in Figure 4 refers to the time each algorithm requires to compute a result for a specific problem, which corresponds to inference time for pre-trained methods. Thus, Figure 4 provides the inference time for "End-to-end Symbolic Regression with Transformers," and Table 9 reports the inference time for "Neural Symbolic Regression that Scales." We clarified this terminology in the revised paper in the caption of Figure 4 and Table 9. We apologize for any confusion regarding the reported times.
> >
> > [1] Brenden K. Petersen, Mikel Landajuela, T. Nathan Mundhenk, Cl´audio Prata Santiago, Sookyung Kim, and Joanne Taery Kim. Deep symbolic regression: Recovering mathematical expressions from data via risk-seeking policy gradients. In 9th International Conference on Learning Representations, ICLR 2021.

---

> > > ### Author Response · Authors · 2024-11-24
> > >
> > > __Noise experiments__
> > >
> > > The noise experiments for ParFam, PySR, and DL-ParFam are now complete, and we've updated Figures 4, 9, 10, and 11. The results show that ParFam maintains the highest accuracy even with highly noisy data but, like most algorithms, struggles to recover the exact equation, often adding small polynomial terms. The performance gap in symbolic solution rates between ParFam and PySR remains similar, though PySR's accuracy declines more significantly (see Figure 4).
> > >
> > > The results for uDSR and EndToEnd follow soon.

---

> > ### Comment · Reviewer_ZXN3 · 2024-11-25
> > **Reply for Authors**
> >
> > Dear Author, thank you very much for your reply. I will ask you the following questions in response to your reply.
> >
> > 1. Using pre-training to predict architecture. Similar to this post. https://doi.org/10.1016/j.neunet.2023.06.046
> >
> > the first contribution of this paper is: 'translating the discrete optimization problem into a continuous one'. I still insist that this is not ParFam's contribution, let alone the main contribution of this article.
> >
> > In short, although the experiment in this paper is quite sufficient, the innovation still needs to be refined again.

---

> > > ### Author Response · Authors · 2024-11-25
> > >
> > > We are happy to hear that the reviewer is satisfied with our experiments and thank the reviewer for the further discussion. We uploaded a newly revised version.
> > >
> > > __Rectified SymNet__
> > >
> > > Thank you for this reference, which has indeed interesting similarities with DL-ParFam. We included it in Section 2.3 in the following way:
> > >
> > > *"The approach most similar to DL-ParFam is SNR, which uses a pre-trained SET-Transformer to predict a mask for active connections within SymNet—a symbolic neural network similar to EQL. During inference, these predictions are further fine-tuned using RL."*
> > >
> > > Key differences between DL-ParFam and SNR lie in their base methods (ParFam vs. SymNet) and architectural encodings (model parameters vs. active connections). The model-parameter encoding in DL-ParFam offers several advantages. It ensures unique labeling, meaning there is exactly one correct label for each problem. It is also dimension-agnostic, allowing the same network to handle any problem dimension up to a preset limit. Additionally, DL-ParFam covers many functions within a single prediction by targeting parametric families, often requiring just three predictions to identify the correct formula.
> > >
> > > In contrast, the advantages of SNR’s active-connection encoding are that it predicts specific functions, which simplifies and accelerates the subsequent optimization process. This specificity makes the SET-Transformer more practically reusable, and fine-tuning it with reinforcement learning (as done in the paper) is more feasible.
> > >
> > > Unfortunately, due to the lack of available SNR code, we cannot provide experimental comparisons.
> > >
> > > __Main contribution__
> > >
> > > We agree that our innovation does not lie in being the first one to translate SR into a continuous optimization problem, but introducing a novel translation of symbolic regression into a continuous optimization problem, making it a competitive alternative to genetic programming-based methods. We will further specify that in the contributions by substituting the first contribution by:
> > >
> > > *"Introduction of ParFam, a novel SR method that improves performance __over existing continuous optimization-based SR algorithms__. ParFam leverages the inherent structure of physical formulas and the expressivity of rational functions to translate SR into an efficiently solvable continuous optimization problem, by avoiding the need for nested basis functions. This results in the following advantages: (1) Enabling gradient-based optimization techniques while avoiding exploding gradients, (2) enhanced interpretability, and (3) efficient but simple and user-friendly setup."*
> > >
> > > We hope that this helps to clears your concerns and are happy to have further discussions.

---

> ### Author Response · Authors · 2024-11-27
>
> Dear Reviewer,
>
> we just wanted to let you know that the noise experiments for uDSR and End2End are also finished now. You can find them in our latest revision.

---

### Official Review · Reviewer_JBKY · 2024-11-03

**Soundness:** 4
**Presentation:** 3
**Contribution:** 3
**Rating:** 8
**Confidence:** 4

**Summary:**

The authors propose a SR method that fixes a rational function structure and optimizes it using MC-based optimization methods. They also present a version of this technique that uses pre-trained transformers as a starting point. Results are presented on many SR benchmark problems and include an analysis of expressivity.

**Strengths:**

The paper is well written and to my knowledge a fairly novel approach to SR that is well-contextualized. The authors have made very, very extensive experimental comparisons, although many of these are sent to the appendix. The authors also do a good job of providing a slightly deeper theoretical justification for their work than in typical SR papers which I enjoyed reading. I think the paper makes a good contribution to the field in showing another avenue for SR that revisits established global optimizers with a unique functional structure that can still find ground truth solutions with good fidelity.

**Weaknesses:**

The main weakness of Feynman and Strogatz dataset comparisons is that many of the equation forms are very simple (especially feynman). So, one way to do well on them is to restrict the complexity of models during optimization (or invent a method that happens to search over simple forms). That's why it's important to also consider benchmarks on real world data. The authors are aware of this, but focus the main body of their text on these synthetic datasets and send real-world results to the appendix. I would be in favor of the real-world dataset comparisons from SRBench being more of a main result, especially if they are given an extra page in the revisions.

I would have also liked to see a stronger connection made between the expressive restrictions of their methods (e.g., inability to model deep unary functions) and the distribution of those types of expressions contained in those benchmarks.

- i would have liked to see more motivation for the chosen representation of equations as rational functions earlier on. Why are shallow rational functions a good _hypothetical_ choice for representing any possible expression tree, before introducing your approach to measuring expressivity.

- typically theorems are self contained. So in Theorem 2.1 it would be more clear to define $c$, $x$, $l$, etc. rather than having to find them throughout the text.

- I found S 2.2 on the expressivity of ParFarm interesting, but I had trouble extracting an intuition for how expressivity scales from the description and from Table 1. It looks like $x$ is being reused for two different variables, and it also appears that the authors are only considering expressivity for $\ell = 1$ i.e. equations with a single function. I would have liked to know how expressivity scales as functions grow in complexity ($l$) which seems equally or more important than scaling by the number of leaves ($n$) and unary functions ($k$).

- DL-ParFam seems to be extremely sensitive to small amounts of noise in the SRBench ground truth problems. I would have liked to see this mentioned in the results/discussion.

- misc
  - awkward phrasing: "to avoid the regularization to be circumvented"

**Questions:**

> Typical formulas from the Feynman dataset Udrescu & Tegmark (2020), for instance, have a complexity of around 10.

What does this mean?

Table 1: It wasn't clear to me why parfarm's ability to cover possible expressions would increase as $n$ increased. Is it just because $n$ artificially inflates the proportion of copies of identical function forms with swapped leaves in $d$? Or something else?

---

> ### Author Response · Authors · 2024-11-21
>
> [1/2]
>
> We thank Reviewer JBKY for their detailed review and help in improving our paper. We uploaded a revised version of the paper to follow the ideas and recommendations of the reviewers.
>
> __It's important to also consider benchmarks on real world data. The authors are aware of this, but focus the main body of their text on these synthetic datasets and send real-world results to the appendix. I would be in favor of the real-world dataset comparisons from SRBench being more of a main result, especially if they are given an extra page in the revisions.__
>
> We appreciate the suggestion to move the black-box experiments to the main paper. We agree that these experiments are important and benefit from greater visibility. Unfortunately, from our understanding of the ICLR author guidelines (https://iclr.cc/Conferences/2025/AuthorGuide) it seems like there will not be an additional page available this year. However, we restructured the benchmark section and moved the description of the datasets to Appendix F, such that we could move the black-box experiments to the main paper.
>
> __I would have also liked to see a stronger connection made between the expressive restrictions of their methods (e.g., inability to model deep unary functions) and the distribution of those types of expressions contained in those benchmarks.__
>
> For the black-box experiments, it is not possible to determine expressivity limitations since the true formulas are unknown. However, for the ground-truth datasets, we can quantify this precisely. Among 133 functions, only one function is not representable by ParFam:
>
> - Feynman: I.29.16. $\sqrt{x_1^2+x_2^2-2x_1x_2\cos(\theta_1-\theta_2)}$
>
> Therefore, ParFam failed to recover the correct formula, however, it found another (more complicated) formula that approximates the data with an $R^2$ of $0.9992$.
>
> __I would have liked to see more motivation for the chosen representation of equations as rational functions earlier on. Why are shallow rational functions a good hypothetical choice for representing any possible expression tree, before introducing your approach to measuring expressivity.__
>
> Thank you for raising this important point. The motivation for our architecture is multi-faceted and partially addressed in Section 1.1 (comparison with EQL) and Subsection 2.1 (discussion on the number of layers). The primary reason for choosing rational layers over linear layers is that they allow ParFam to achieve strong approximation capabilities with only one hidden layer, due to the high approximation qualities of rational functions and the general structure of common physical laws. The advantages of only having one hidden layer are:
>
> - **Low dimensionality**
>
> - **Ease of optimization**, as it reduces issues like exploding/vanishing gradients
>
> - **Flexibility** to incorporate additional basis functions, e.g., the exponential would cause exploding gradients for multiple layers
>
> - **Enhanced interpretability**, as nested unary functions can be harder to understand
>
> - **Aligns with the structure of physical laws**
>
> We recognize that this motivation may not have been sufficiently clear in our original presentation and revised the last paragraph in Section 2.1.1 to make this point more explicit.
>
> __It would be preferable if Theorem 1 would be self-contained__
>
> Thank you for pointing this out, we adapted it in the revised version accordingly for Theorem 2.1 and 2.2 as well.

---

> > ### Author Response · Authors · 2024-11-21
> >
> > [2/2]
> >
> > __Clarifying the expressivity of ParFam (Section 2.2)__
> >
> > We apologize for any confusion caused in this section. An intuition for the expressivity can be derived from Table 1, which shows $\frac{r_2}{x_1}$  for different values for $k$ and $n$. This ratio is approximately equal to $\frac{c_{l+1}/c_l}{d_{l+1}/d_l}$. Since $c_0=d_0$ holds, we can compute $\frac{c_l}{d_l}\approx(\frac{r_2}{x_1})^l$.
> >
> > For example, with $n=4$ and $k=3$, Table 1 gives us $\frac{r_2}{x_1}=0.9799$. Therefore, $\frac{c_l}{d_l}=0.9799^l$. For $l=5$, ParFam covers ~90.25% of formulas, and for $l=10$, ~81.62%.
> >
> > We hope that this helps understanding the theoretical part better and welcome any further recommendations on improving its accessibility. We added the above example in the end of the section in the revised version.
> >
> > __DL-ParFam seems to be extremely sensitive to small amounts of noise in the SRBench ground truth problems. I would have liked to see this mentioned in the results/discussion.__
> >
> > We acknowledge that DL-ParFam shows sensitivity to small amounts of noise in the SRBench ground-truth problems and agree that this observation should be included in the results and discussion. We address this in the revised manuscript by adding:
> >
> > "However, DL-ParFam's ability to recover the symbolic solution is notably hindered under low-noise conditions."
> >
> > __awkward phrasing: "to avoid the regularization to be circumvented"__
> >
> > Thank you for noting this. We rephrased this in the revised version.
> >
> > __"Typical formulas from the Feynman dataset Udrescu & Tegmark (2020), for instance, have a complexity of around 10." What does this mean?__
> >
> > Our intent was to provide a sense of the typical complexity of formulas in the Feynman dataset. For example, a formula such as $m\sin(n\theta/2)^2/\sin(\theta/2)^2$ (Feynman I.30.3) has a complexity of 9, measured by the number of non-leaf nodes in its expression tree. However, we agree that the sentence as written may be confusing. We added the above example as an explanation but are also open to either rephrasing or removing it if the reviewer finds it more appropriate.
> >
> > __Table 1: It wasn't clear to me why parfarm's ability to cover possible expressions would increase as $n$ increased. Is it just because artificially inflates the proportion of copies of identical function forms with swapped leaves in $d$? Or something else?__
> >
> > This is indeed an interesting and nuanced phenomenon. The mathematical explanation aligns with your observation: increasing $n$ inflates the proportion of function forms with swapped leaves. Specifically, as the number of binary nodes in an expression tree increases, the number of leaves also increases. Consequently, trees with more binary nodes constitute a larger proportion of the space of unary-binary trees for higher $n$.
> >
> > Since ParFam is better at handling binary operators than unary ones, the formulas it covers tend to have more binary operators (and therefore more leaves) on average. This leads to an increase in the coverage ratio as $n$ grows.
> >
> > While this may seem theoretical, it aligns with practical intuition: a "simple" formula involving many variables often needs to be shallow, favoring binary operators over unary ones, which aligns with ParFam's strengths.

---

### Meta-Review · Area_Chair_UVkc · 2024-12-20

**Metareview:**

The paper considers the problem of symbolic regression and provides a new approach that translates discrete optimization into continuous optimization by using parametric families of functions. The paper provides a careful analysis of the resulting optimization problem and how to solve it with a global-local basin hopping algorithm. The expressivity of the function class is discussed and an extension (DL-ParFam) is proposed that uses a pre-trained transformer to guide parameter selection. The method is evaluated on standard benchmarks in symbolic regression and show excellent performance.

Overall, I found the approach to be well motivated and carefully analyzed in the paper. I also liked that the supplementary available code seems easy to use which will be helpful for the community. Theoretical analysis of expressivity is useful for descriptive study of the approach. Two reviewers JBKY and 3uKW liked the paper and although reviewer ZXN3 gave a reject, I felt it was unfair given that the authors clarified most of their concerns. Therefore, I recommend acceptance with request to the authors to update the papers based on discussion in the rebuttal period.

**Additional Comments On Reviewer Discussion:**

Reviewer JBKY praised the paper's well-written nature and novel approach, extensive experimental comparisons and valued the theoretical justification provided. Their main concern was over-focus on synthetic datasets (Feynman/Strogatz) versus real-world data. Authors response acknowledged the importance of real-world benchmarks and restructured the paper to emphasize it. Reviewer ZXN3 questioned the novelty of continuous optimization formulation of symbolic regression, requested comparison with baselines and asked about specific hyperparameter settings. Authors responded clearly that this is not the claim and also added new experiments which was satisfactory in my opinion. However, reviewer ZXN3 didn't acknowledge this and therefore I down weighted their review of lower score. Reviewer 3uKW appreciated theoretical analysis and benchmark performance.

---

### Decision · Program_Chairs · 2025-01-22

Accept (Poster)